# Integrative functional genomics decodes herpes simplex virus 1

Adam W. Whisnant [1,10], Christopher S. Jürges [1,10], Thomas Hennig[1], Emanuel Wyler [2], Bhupesh Prusty [1], Andrzej J. Rutkowski[3], Anne L'hernault[3], Lara Djakovic [1], Margarete Göbel[4], Kristina Döring[4], Jennifer Menegatti[5], Robin Antrobus[6], Nicholas J. Matheson [6], Florian W. H. Künzig [1], Guido Mastrobuoni [2], Chris Bielow [2], Stefan Kempa [2], Chunguang Liang [7], Thomas Dandekar [7], Ralf Zimmer[8], Markus Landthaler [2], Friedrich Grässer[5], Paul J. Lehner[6], Caroline C. Friedel [8], Florian Erhard [1✉] & Lars Dölken [1,3,9✉]

The predicted 80 open reading frames (ORFs) of herpes simplex virus 1 (HSV-1) have been intensively studied for decades. Here, we unravel the complete viral transcriptome and translatome during lytic infection with base-pair resolution by computational integration of multi-omics data. We identify a total of 201 transcripts and 284 ORFs including all known and 46 novel large ORFs. This includes a so far unknown ORF in the locus deleted in the FDA-approved oncolytic virus Imlygic. Multiple transcript isoforms expressed from individual gene loci explain translation of the vast majority of ORFs as well as N-terminal extensions (NTEs) and truncations. We show that NTEs with non-canonical start codons govern the subcellular protein localization and packaging of key viral regulators and structural proteins. We extend the current nomenclature to include all viral gene products and provide a genome browser that visualizes all the obtained data from whole genome to single-nucleotide resolution.

[1] Institute for Virology and Immunobiology, Julius-Maximilians-University Würzburg, Versbacher Straße 7, 97078 Würzburg, Germany. [2] Berlin Institute for Medical Systems Biology, Max-Delbrück-Center for Molecular Medicine, 13125 Berlin, Germany. [3] Department of Medicine, University of Cambridge, Box 157, Addenbrookes Hospital, Hills Road, CB2 0QQ Cambridge, UK. [4] Core Unit Systems Medicine, Julius-Maximilians-University Würzburg, Josef-Schneider-Str. 2/D15, 97080 Würzburg, Germany. [5] Institute of Virology, Building 47, Saarland University Medical School, 66421 Homburg, Saar, Germany. [6] Cambridge Institute of Therapeutic Immunology & Infectious Disease (CITIID), Department of Medicine, Cambridge Biomedical Campus, University of Cambridge, Puddicombe Way, Cambridge CB2 0AW, UK. [7] Department of Bioinformatics, Biocenter, Am Hubland, Julius-Maximilians-University Würzburg, 97074 Würzburg, Germany. [8] Institute of Informatics, Ludwig-Maximilians-Universität München, Amalienstr. 17, 80333 Munich, Germany. [9] Helmholtz Institute for RNA-based Infection Research (HIRI), Helmholtz-Center for Infection Research (HZI), 97080 Würzburg, Germany. [10] These authors contributed equally: Adam W. Whisnant, Christopher S. Jürges. ✉email: florian.erhard@uni-wuerzburg.de; lars.doelken@uni-wuerzburg.de

Herpes simplex virus 1 (HSV-1) is the causative agent of the common cold sores but also responsible for severe, life-threatening disease including generalized skin infections, pneumonia, hepatitis, and encephalitis[1]. The HSV-1 genome is about 152 kb in size and known to encode at least 80 open reading frames (ORFs), many of which have been extensively studied. Large-scale RNA-seq and ribosome profiling recently revealed that the coding capacity of three other herpesviruses, namely human cytomegalovirus (HCMV), Kaposi's sarcoma-associated herpesvirus (KSHV) and Epstein-Barr Virus (EBV) is significantly larger than previously thought[2–5]. For HCMV and KSHV in particular, hundreds of viral gene products were identified. These result from extensively regulated usage of alternative transcription and translation start sites throughout lytic infection. Moreover, these viruses were found to encode hundreds of short ORFs (sORFs) of unknown function. Similar to their cellular counterparts, these may either regulate translation of viral gene products or encode for functional viral polypeptides[6–10]. To date, the majority of these viral gene products have not been experimentally validated. Furthermore, the lack of a complete annotation and a revised nomenclature severely hampers functional studies.

Here, we employed a broad spectrum of unbiased functional genomics approaches and reanalyzed recently published data to comprehensively characterize HSV-1 gene products (Fig. 1). Our analysis of the viral transcriptome included: previously published time-course experiments of (i) total RNA-seq and 4sU-seq data[11], (ii) unpublished transcription start site (TiSS) profiling using two complementary approaches (cRNA-seq[2] and dRNA-seq[12]), (iii) incorporation of viral transcripts recently identified by two other groups using PacBio[13] and MinION[14] platforms, and (iv) RNA localization by RNA-seq of subcellular fractions of both wild-type HSV-1[15] and the deletion mutant of the key viral RNA export factor ICP27. Analysis of the viral translatome included: (i) standard ribosome profiling[11] as well as so far unpublished translation start site (TaSS) profiling using (ii) Harringtonine and (iii) Lactimidomycin. Viral ORFs were validated using whole-cell quantitative proteomics and reverse genetics approaches. To make the annotation and all the obtained data readily accessible to the research community, we provide an in-house genome browser software tailored to the visualization of HSV-1 and our collection of data (available at http://software.erhard-lab.de) as well as all data files to browse our annotation and data with any available genome browser at Zenodo[16]. Thereby, viral gene expression and all data can be visually examined from whole genome to single-nucleotide resolution.

In total, we expanded the number of known of HSV-1 genomic elements to 201 viral transcripts encoding a total of 284 ORFs; including N-terminal peptide extensions and truncations of several classically described viral proteins as well as previously unannotated protein-coding sequences in the loci of genes for major regulatory proteins ICP0 and ICP34.5.

## Results

**Characterization of the HSV-1 transcriptome.** To identify the full complement of viral transcripts, we performed TiSS profiling employing a modified RNA sequencing protocol that is based on circularization of RNA fragments (here termed cRNA-seq)[2]. It enables quantification of RNA levels as well as identification of 5′ transcript ends by generating a strong enrichment (≈18-fold) of reads that start at the 5′ RNA ends. With cRNA-seq, we identified 266 potential TiSS (see Supplementary Methods criteria ii–iv) that explained the expression of many previously annotated viral coding sequences (CDS). To comprehensively identify and validate putative TiSS, we applied a second 5′-end sequencing

approach termed differential RNA-seq (dRNA-seq)[12], which provides a much stronger (≈300-fold) enrichment of TiSS at increased sensitivity (446 potential TiSS, see Supplementary Methods criterion i). It is based on selective cloning and sequencing of the 5′-ends of cap-protected RNA molecules that are resistant to the 5′–3′-exonuclease Xrn1. The two approaches provided highly consistent data at single-nucleotide resolution (Fig. 2a). Furthermore, we reassessed viral transcripts called by two other groups exclusively based on third-generation sequencing techniques (MinION[14] and PacBio[13] platforms, see Supplementary Methods criteria v and vi). This confirmed many of our TiSS (Supplementary Fig. 1a,b). The 80 viral transcripts (corresponding to a total of 89 TiSS, some of which were only separated by 5 nt), which were recently identified using MinION data, generally lacked 7-18 nucleotides (nt) at the 5′ end due to technical limitations of the MinION direct RNA sequencing method (Supplementary Fig. 1b)[17]. After correcting this bias using our data, MinION-derived TiSS were highly consistent with our cRNA-seq and dRNA-seq data (Fig. 2b). Only 11 of the 89 TiSS (12%) identified by Depledge et al. could not be confirmed. We thus did not adopt them into our final genome annotation. Nevertheless, our genome browser encodes a separate track that visualizes all MinION and PacBio transcripts. Around half of all the TiSS that were previously identified using PacBio sequencing[13] matched to our data with single-nucleotide resolution. The remaining TiSS (108 of 201; 54%) could neither be confirmed by cRNA-seq nor dRNA-seq (Fig. 2c). Most of them were only called from very few reads and presumably represent cleavage products of larger viral RNAs. This demonstrates that complementary experiments are essential to exclude false positives and that none of the approaches by themselves is sufficient to reliably identify all viral TiSS.

To comprehensively assess the viral TiSS candidates that were only identified by a single approach, we developed a computational pipeline termed iTiSS (integrative Transcriptional Start Site caller). It screens potential TiSS for clustered accumulation of read 5′-ends in dRNA-seq (i) and cRNA-seq (ii) data. It evaluates our cRNA-seq data for an increase in upstream to downstream read coverage at potential TiSS (iii), and temporal changes in the potential TiSS read clusters in the cRNAs-seq time-course data (iv). It accounts for TiSS already identified by MinION (v) and PacBio (vi) sequencing. In addition, we also analyzed our 4sU-seq time-course data to both score potential TiSS that explained temporal changes in expression levels throughout infection (vii), and an increase in upstream to downstream read coverage (viii). Finally, we also scored TiSS that explained translation of viral ORFs, for which no other transcript had otherwise been identified (ix). For more details on each criterion see Supplementary Methods. Thus, 9 criteria were utilized to confirm a potential TiSS. All identified TiSS were manually assessed and curated. In total, this resulted in 189 bona fide viral TiSS, of which 161 (85%) were called by at least 2 criteria (Fig. 2d). Three of the five transcripts (LAT[18], AL-RNA[19], and US5.1[20]), which we could not confirm by any method, had previously been convincingly validated by other groups and were thus included. The other two were included after careful manual inspection (see Supplementary Methods). The complete set of HSV-1 transcripts with their respective scores is provided in Supplementary Data 1.

TATA-boxes are a key element of eukaryotic promoters located 25–30 bp upstream of the TiSS[21]. They are also prevalent for herpesvirus genes[22]. We used our dRNA-seq data to categorize viral RNAs into three transcription categories (high, mid and low transcription). A TATA-box or TATA-box-like motif was found significantly more often in the highly transcribed category than in the lowly transcribed one ($p < 10^{-6}$, Fisher's exact test). In mammalian cells, the TiSS is marked by the initiator element

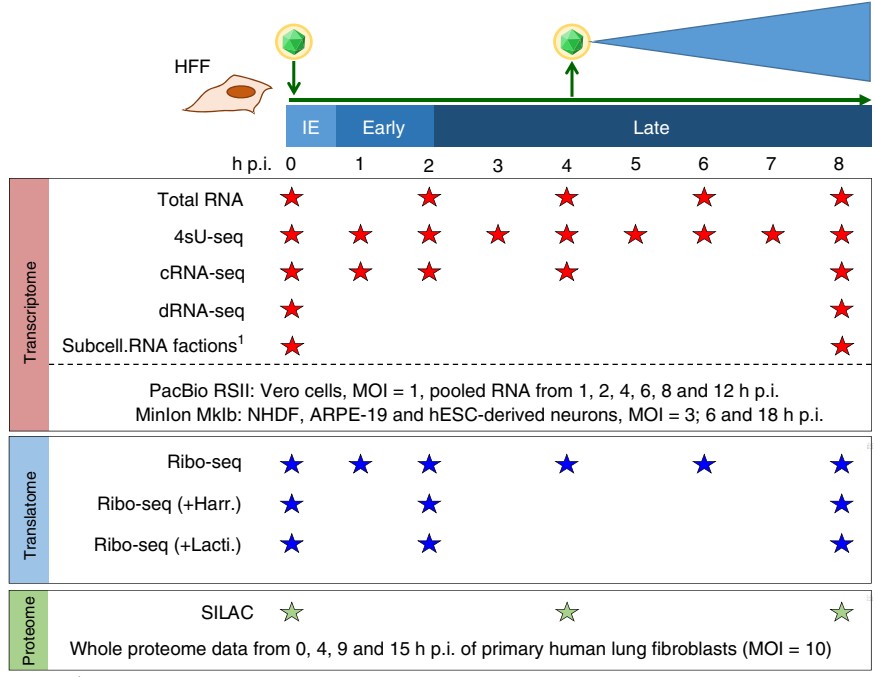

**Fig. 1 Overview of the applied Omics approaches.** Viral gene expression was analyzed in primary human fibroblasts (HFF). The total RNA-seq, 4sU-seq and ribosome profiling data were recently published[11]. To comprehensively identify transcription start site (TiSS), we performed cRNA-seq[2] and dRNA-seq[12] as well as RNA-seq on subcellular RNA fractions from mock, wild-type and ΔICP27-infected cells. For all of these, two biological replicates were performed. Furthermore, we incorporated the recently published transcripts originating from PacBio[13] and MinION[14] sequencing data. Translation start site (TaSS) profiling was performed by ribosome profiling following treatment of cells for 30 min with either Harringtonine or Lactimidomycin[2]. Proteome analysis included two whole-proteome datasets using SILAC and label-free mass spectrometry. The available timepoints and conditions are indicated by stars.

(Inr), the core of which is a pyrimidine-purine (PyPu) dinucleotide[23]. Interestingly, PyPu was also prevalent for the viral TiSS independent of expression levels (Fig. 2e). This provides strong evidence for the TiSS of even the most weakly expressed viral transcripts.

We next looked at splicing within the HSV-1 transcriptome based on our total RNA-seq and 4sU-seq data[11]. We first identified all unique reads that spanned putative exon-exon junctions by at least 10 nt. This confirmed all 8 well-described splicing events and identified an additional tandem acceptor site (NAGNAG)[24] for both the third exon of the ICP0 gene (RL2) and the UL36.6 gene. Recently, Tombácz et al. proposed 11 splicing events based on PacBio sequencing data[13]. Our data confirmed all of these splicing events. However, only 4 of them occurred at relevant levels compared to the overall transcript levels (Supplementary Data 2). Two of these explained translation of small ORFs (UL40.5 iORF and UL40.7 dORF). Finally, we identified 44 so far unknown putative splicing event sites based on our Illumina data (Supplementary Fig. 2 and Supplementary Data 2). However, all of these showed substantially lower read coverage than the surrounding exons, indicating that they only represented rare events at best. Therefore, we decided not to include these low abundance splicing events into our final reference annotation.

A recent paper by Tang et al.[25] proposed 71 novel HSV-1 splicing events. We also observed 15 of these in our Illumina data. Interestingly, about half (28 of 71) of their splicing events reported were exclusively observed upon infection with an ICP27 null mutant. Of note, none of our 44 putative splicing events were found to be more abundant upon infection with an ICP27-null mutant (subcellular RNA fractions from ΔICP27-infected cells). We conclude that they do not reflect aberrant splicing events that

originate from infected cells, which express insufficient levels of ICP27. In total, we thus identified 189 viral TiSS that give rise to at least 201 transcripts and transcript isoforms.

**RNA 3′-end processing and export of viral transcripts.** Previous studies reported regulated usage of the 47 viral poly(A) sites during productive infection, which appeared to be mediated or at least influenced by the viral ICP27 protein[26–31]. We recently reported that lytic HSV-1 infection results in a widespread but nevertheless selective disruption of transcription termination of host genes[11]. In contrast to the extensive read-through transcription at host poly(A) sites that we observed by 4–8 h p.i., viral gene expression remained mostly unaffected. Recently published third-generation sequencing data proposed numerous very large viral transcripts spanning multiple viral genes[14]. To address the nature of these transcripts and their role in translation, we performed RNA-seq on subcellular RNA fractions (total RNA, cytoplasmic RNA, nucleoplasmic RNA and chromatin-associated RNA) using both wild-type HSV-1 and a null mutant of the viral RNA export factor ICP27 (ΔICP27). The data from wild-type HSV-1 and mock infected cells were recently published[15]. The ΔICP27 infection had been performed in the same experiment. Consistent with the well-characterized role of ICP27 in viral mRNA export[32], all viral transcripts appeared to be more efficiently (≈11-fold) exported to the cytoplasm in wild-type than in ΔICP27 HSV-1 infection (Fig. 2f). Interestingly, this even included the spliced immediate early (IE) genes ICP0 (≈5-fold), ICP22 (≈17-fold) and ICP47 (≈27-fold) as well as the unspliced (IE) ICP4 gene (≈11-fold). In chromatin-associated, nuclear and total cellular RNA, considerable numbers of reads were observed within the first 500 nt downstream of viral poly(A) sites (PAS).

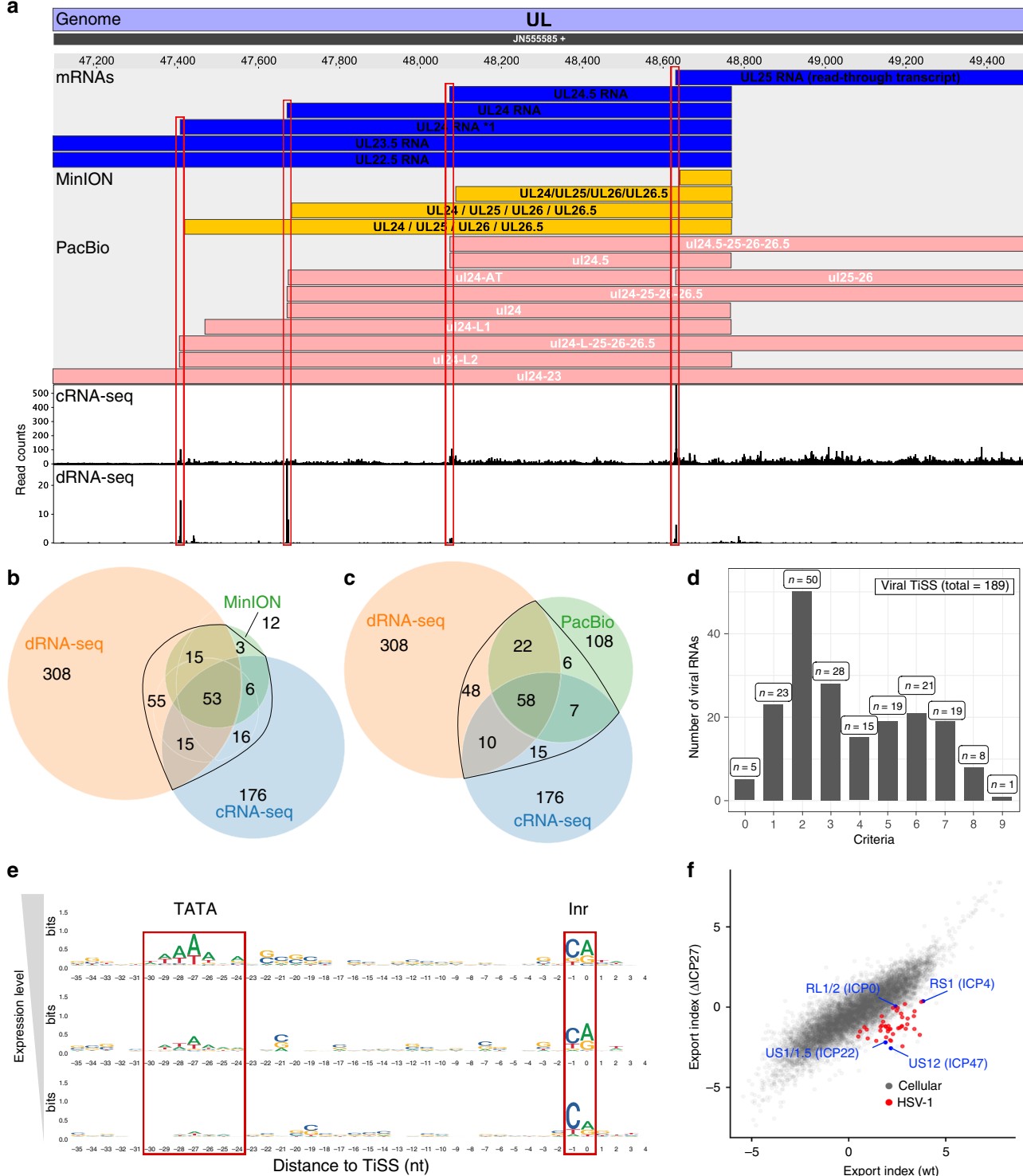

**Fig. 2 Analysis of viral transcription start sites (TiSS). a** Screenshot of our HSV-1 viewer displaying the annotated mRNAs of MinION, PacBio and coverage of read 5′-ends for cRNA-seq and dRNA-seq of the UL22.5-UL25 gene locus. Transcripts in our reference annotation are indicated in blue. **b**, **c** Venn diagram depicting the number of Transcription start sites (TiSS) that were identified by cRNA-seq, dRNA-seq and MinION[14] (**b**) or PacBio[13] (**c**) sequencing, respectively. TiSS included into the final annotation are indicated by the black circle. **d** Histogram depicting the number of TiSS criteria that were fulfilled by the individual viral transcripts. **e** Sequence logos upstream of viral TiSS with viral TiSS grouped into three equally sized bins according to their transcription rates (top: highest; bottom: lowest). The TATA-box and initiator element (Inr) are shown. **f** Log fold-change between cytoplasmic and chromatin-associated FPKM-normalized read counts (export index) of cellular (gray) and viral (red and blue) gene clusters compared between wild-type HSV-1 (wt) and a null mutant of the viral RNA export factor ICP27 (ΔICP27). Viral immediate early genes are indicated in blue.

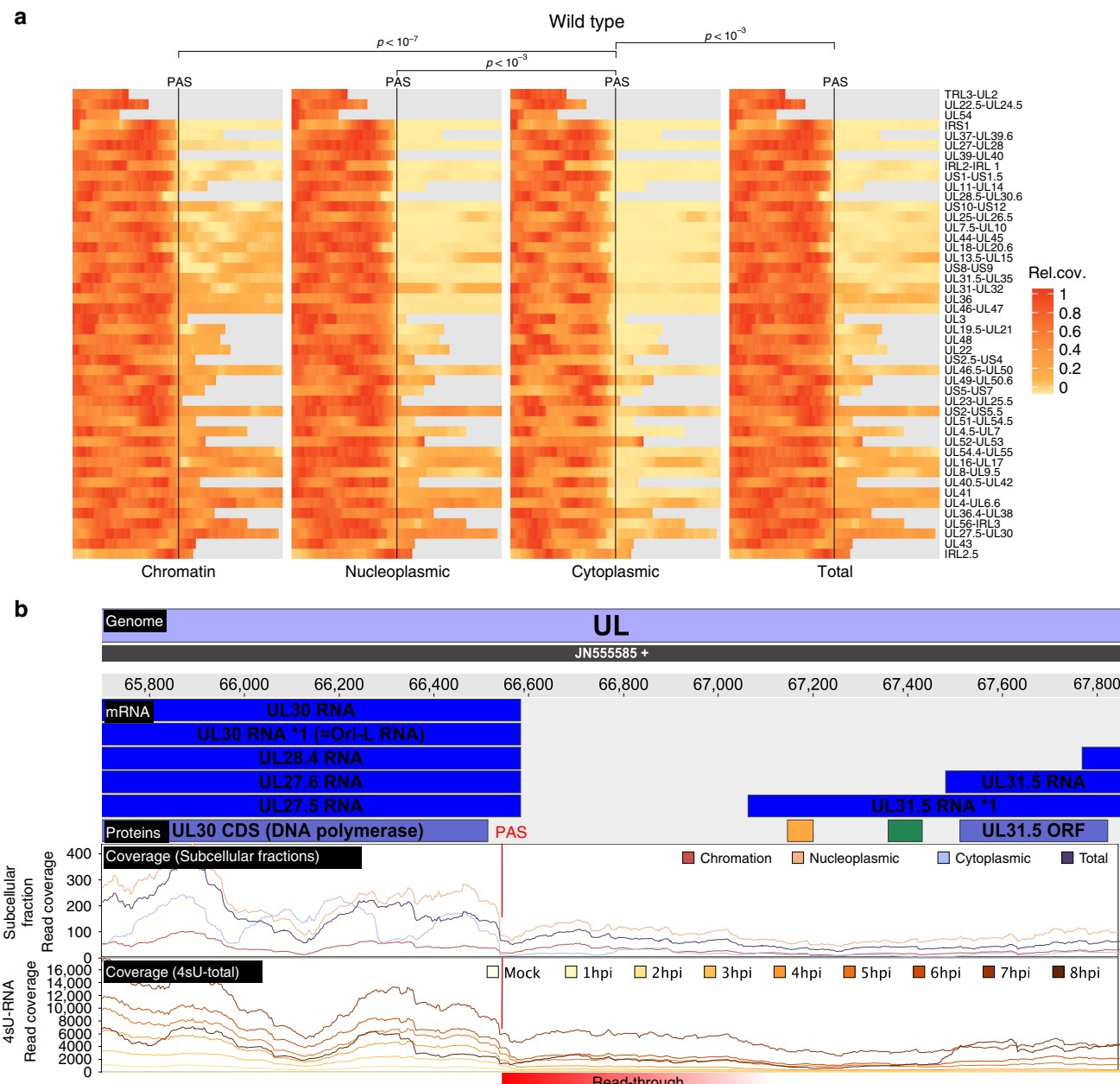

**Fig. 3 Subcellular localization of viral transcripts. a** Read levels 500 bp upstream (left) and downstream (right) of the Poly(A)-site (PAS) of viral genes for wild-type HSV-1. Gray bars indicate overlapping parts with other genes, for which reads could not be uniquely assigned. In the cytoplasmic RNA fraction, read levels dropped substantially immediately downstream of the PAS. p-values were calculated using a one-sided paired t-test over the mean fold-change of read levels 500 bp before against after the PAS (Cytoplasmic to Nucleoplasmic: $p$-value $= 8.157 \times 10^{-4}$, Cytoplasmic to chromatin-associated: $p$-value $= 6.956 \times 10^{-8}$, Cytoplasmic to total $p$-value $= 4.06 \times 10^{-4}$). **b** Screenshot of our HSV-1 viewer depicting poly(A) read-through at the UL30 PAS in cRNA-seq data at 2, 4, and 8 h post infection (hpi) of replicate 1. The annotated transcripts (mRNA), proteins (Proteins) and read coverage (Coverage) for chromatin-associated, nucleoplasmic, cytoplasmic and total reads are shown for the positive strand only. Read-through transcription is schematically indicated in red. Downstream of the UL30 PAS, chromatin-associated and nucleoplasmic reads show substantial read-through, whereas cytoplasmic reads drop down to only a fraction of what they were before (blue arrow).

However, in the cytoplasmic RNA fraction of infected cells, read levels dropped substantially immediately downstream of the PAS (Fig. 3a). This indicates that reads mapping downstream of the PAS reflect mRNA precursors, which remain nuclear and, thus, do not contribute to the translated viral transcriptome. However, for some viral genes, e.g., *UL30*, *UL38*, and *UL43*, considerable numbers of reads that mapped downstream of the respective PAS were present in the cytoplasmic RNA fraction. For the *UL30* PAS,

this became substantially more prominent late in infection (8 h p. i., Fig. 3b). Furthermore, transcription of *UL25*, which initiates 107 nt upstream of the *UL24* PAS, efficiently bypassed the *UL24* PAS already from 2 h p.i. on (Supplementary Fig. 3). The same was observed for *UL24.5* which represents an N-terminal truncated isoform of UL24. These data confirm previous findings on differential polyadenylation of selected viral genes during productive infection[26–30].

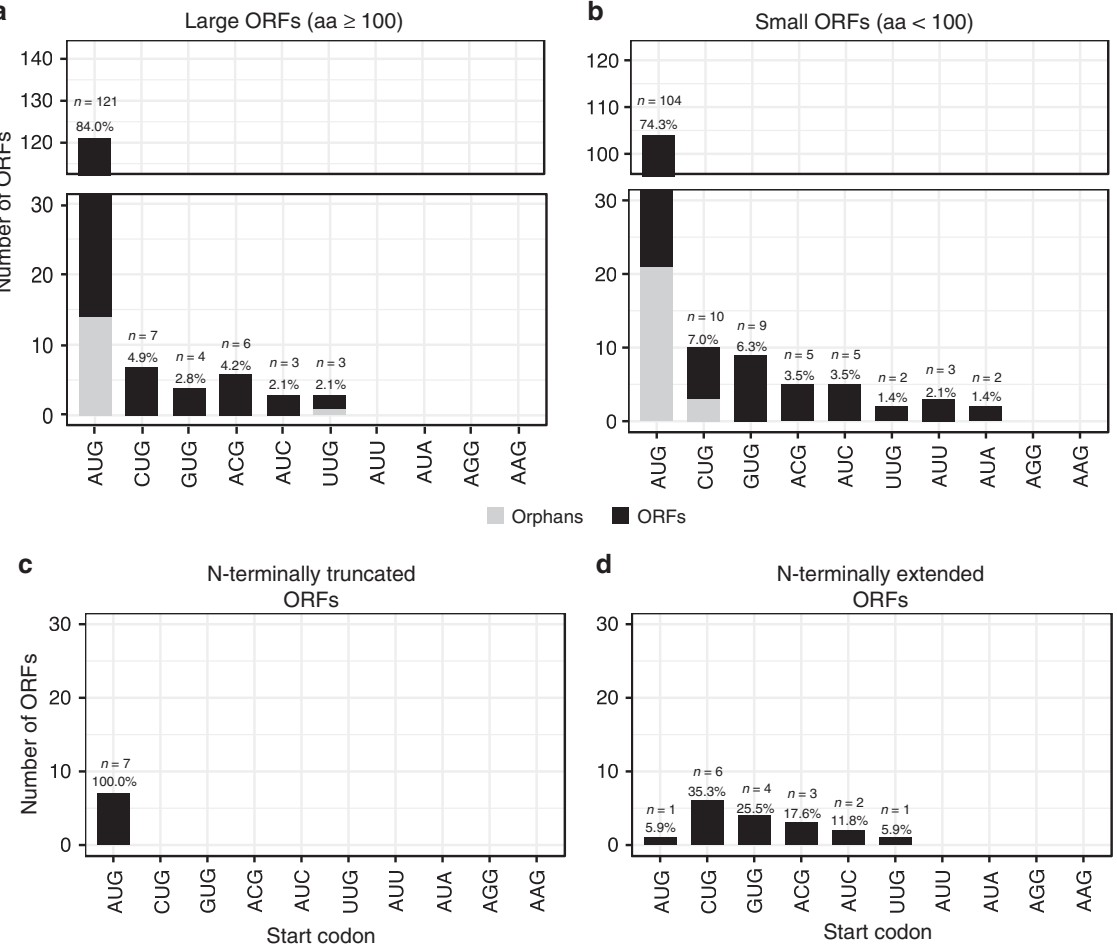

**Fig. 4 Distribution of start codon usage of all identified HSV-1 proteins.** Distribution and frequency of possible start codons used by HSV-1 open reading frames (ORFs) (**a**), short ORFs (sORFs) (**b**), N-terminal truncated ORFs (NTTs) (**c**), and N-terminal extended ORFs (NTEs) (**d**). Orphan ORFs are depicted in light grey. Six of the previously identified CDS (UL11, UL49.5, US5, US9, US12, and RL2 iso1) are <100 aa and were thus included in **b**.

**HSV-1 expresses hundreds of so far unknown ORFs and sORFs.** To comprehensively identify the viral translatome, we performed time-course analysis of ribosome profiling as well as translation start site (TaSS) profiling (see Fig. 1 and Supplementary Methods). The obtained data confirmed the expression of all 80 previously annotated ORFs (CDS) and detected 46 additional large ORFs and 134 small ORFs (3–99 aa). We also identified 7 N-terminal truncations (NTTs) and 17 N-terminal extensions (NTEs) of CDS. In total, our data provides evidence for the translation of 284 viral ORFs (Supplementary Data 3 and 4). Translation predominantly initiated from AUG start codons (79%). However, non-canonical initiation events also substantially contributed to the HSV-1 translatome with CUG, GUG, ACG, and AUC together initiating translation of about 15% and 20% of all large and small viral ORFs, respectively (Fig. 4a, b).

We observed seven NTTs originating from downstream translation initiation events of previously described viral coding sequences (Supplementary Data 5). All of these initiated with AUG start codons (Fig. 4c). Alternative TiSS downstream of the main TiSS explained translation of 6 of 7 NTTs (Supplementary Fig. 4). Only for US3.5, we could not identify a corresponding transcript. It thus remains unclear whether US3.5 is translated from an independent transcript or due to leaky scanning. Six of these NTTs (UL8.5, UL12.5, UL24.5, UL26.5, US1.5, and US3.5) had already been reported[33–38]. Only the NTT of the major

DNA-binding protein pUL29 (ICP8; comprising aa 516–1212) had so far not been described. Interestingly, this NTT initiates at an AUG start codon immediately downstream of the metal-(Zn)-binding loop (residues 499–512)[39,40]. While the ribosome profiling data showed a strong peak at the respective AUG start codon, which was further enriched by LTM treatment (Supplementary Fig. 4), we were unable to validate the truncated UL29.5 protein using a C-terminally 3×-Flag-tagged mutant virus. Further experiments are thus required to clarify the existence and stability of UL29.5 as well as its corresponding transcript.

Interestingly, 16 of the 80 viral reference ORFs (20%) showed in-frame NTEs (Supplementary Data 6) with translational activity exceeding 10% of the main downstream ORF. The majority of NTEs (16 of 17, including 2 NTEs in UL50) initiated translation from non-AUG start codons (Fig. 4d). This included key viral proteins like the major immediate early protein ICP27 (UL54), the major capsid protein (VP5, UL19) and the well-studied viral kinase US3. For five viral genes, we generated mutant viruses by introducing a 3×-FLAG-tag either into the NTE or downstream of the canonical AUG start codon. This confirmed the expression of 6 NTEs including the 2 UL50 NTEs (Fig. 5a–e). Interestingly, the introduction of a 3×-FLAG-tag into the N-terminal extension of both ICP27 and VP5 resulted in dead viruses, which could only be reconstituted upon transfection of complementing cells. For UL54, expression of the NTE was already observable when the

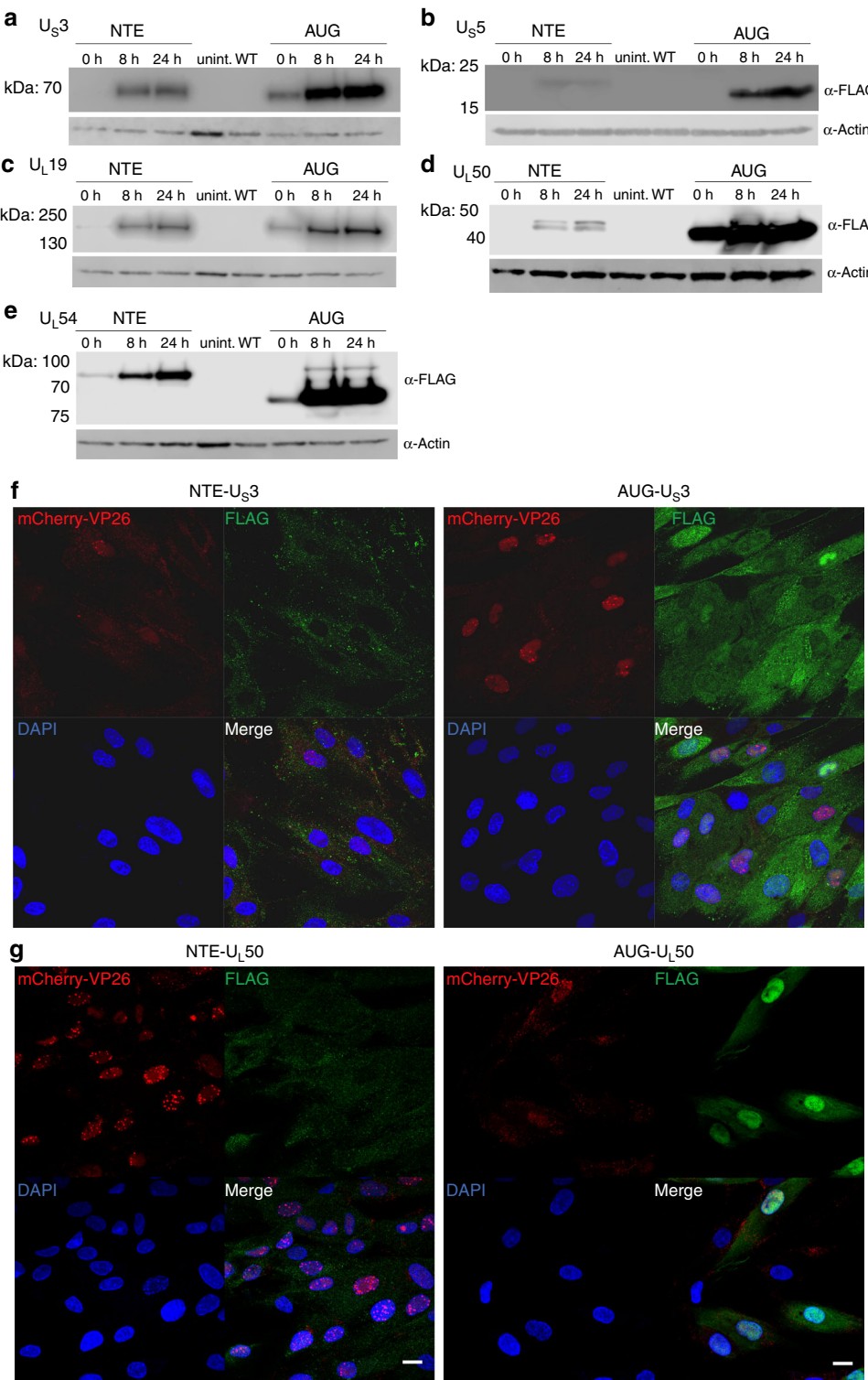

**Fig. 5 Validation of N-terminal extensions of known HSV-1 proteins.** Tagged viruses were generated by inserting a 3×-FLAG-tag either upstream of the canonical start codon into the N-terminal extension (NTE) or downstream of it (AUG). Western blots of 3×-FLAG-tagged N-terminal extensions following infection of human foreskin fibroblasts with the indicated viruses are shown. Expression at the given hours (h) post-infection are compared to uninfected (uninf.) and the parental (WT) virus, both at 24 h p.i. for the HSV-1 genes **a** US3, **b** US5, **c** UL19, **d** UL50, and **e** UL54. Expression of the NTE of UL54 (ICP27) was already visible when the 3×-FLAG-tag was inserted downstream of the canonical AUG. **f** Immunofluorescence of human foreskin fibroblasts infected with mCherry-VP26 HSV-1 containing 3×-FLAG-tags inserted upstream of the canonical start codon into the N-terminal extension (NTE) or downstream of it (AUG) for US3 and **g** UL50. Cell nuclei were stained using DAPI. Scale bars depict 20 microns. Protein localization of both NTEs shifts to the cytoplasm. Source data are provided within the Source Data file.

3×-FLAG-tag was introduced downstream of the canonical AUG start codon (Fig. 5e). For UL19 (VP5) major capsid protein (MCP), the 3×-FLAG-tagged NTE appeared to even be dominant negative. Virus reconstitution in non-complementing cells resulted in a partial deletion of the NTE within two passages. This indicates that the 3×-FLAG-tagged NTE-MCP is assembled into virus particles but renders them dysfunctional due to the N-terminally inserted 3×-FLAG-tag.

To test the impact of the respective NTEs on protein localization, we performed immunofluorescence microscopy of both the NTE- and AUG-tagged viruses. While subcellular localization of the NTEs of UL54 and US5 were indistinguishable from their canonical counterparts (Supplementary Fig. 5), the NTEs of US3 and UL50 notably altered subcellular localization (Fig. 5f, g). While canonical US3 was predominantly nuclear, the NTE-US3 localized to the cytoplasm. The US3 NTE contains a leucine-rich stretch indicating a putative nuclear export signal. Pseudorabies virus (PRV), a porcine alphaherpesvirus, expresses two isoforms of US3, both of which initiate from AUG start codons on separate transcripts (Supplementary Fig. 6). The longer isoform encodes a mitochondrial localization signal resulting in the cytoplasmic localization and a failure of the respective protein to be incorporated into the tegument[41]. The DNA sequence of the US3 NTE is conserved in HSV-2 and its role as a nuclear export signal fits data demonstrating that HSV-2 US3 fails to accumulate in the cytoplasm when nuclear export is inhibited[42]. Similar to US3, localization of NTE-UL50 also shifted to the cytoplasm (Fig. 5g). UL50 dUTPase activity in PRV-infected cells was reported to be nuclear[43], while it appears to be predominantly cytoplasmic with HSV-2[44] and nearly equally distributed in HSV-1. We conclude that NTEs initiating from non-AUG start codons are common in alphaherpesvirus proteomes. They allow the expression of alternative protein isoforms with different subcellular localization and regulatory motifs.

In 2015, the first oncolytic virus (talimogene laherparepvec (Imlygic)) was approved for therapy of melanoma[45]. This modified herpes simplex virus 1 lacks two viral genes (ICP34.5 and ICP47) and expresses GM-CSF to recruit and stimulate antigen-presenting cells. Within the ICP34.5 (RL1) locus, we found a 93 aa ORF, which we termed RL1A (Fig. 6a). It initiates from an AUG start codon 46 nt upstream of the AUG start codon of RL1 and is translated from the same transcript at >4-fold higher levels. Imlygic thus lacks a third viral protein, namely RL1A.

The RL2 locus encodes the major viral immediate early protein ICP0. Here, we identified an additional spliced ORF (termed RL2A) of 181 aa that initiates from an ACG start codon 116 nt upstream of the ICP0 TaSS (Fig. 6b). Expression of RL2A was confirmed by generating a mutant virus (3×-Flag-RL2A) with a 3×-FLAG-tag inserted 12 nt downstream of the ACG start codon (Fig. 6c). Interestingly, RL2A expression by the mutant virus could only briefly be detected immediately upon virus reconstitution and was readily lost upon serial passaging. This indicates that insertion of the 3×-FLAG-tag into the RL2A repeat region severely impaired viral fitness resulting in DNA recombination with the other wild-type repeat. To address this issue, we generated a second mutant virus (3×-FLAG-RL2A-ΔRL) by subsequently deleting the wild-type RL2A and part of RL2 from the second repeat to prevent recombination and removal of the inserted 3×-Flag-tag upon virus reconstitution and passaging. This resulted in stable expression of 3×-Flag-tagged RL2A of the expected size (21.8 kD; Fig. 6c). Interestingly, however, ICP0 expression of this mutant was almost completely abolished. We subsequently noted that the 3×-FLAG-tag contains an out-of-frame AUG start codon (GATTACAAGG**ATG**ACGACGATAA) in every of the three FLAG-tag repeats. Translation initiation at

the respective start codons and ribosomes bypassing the ICP0 TaSS explains the observed near-complete loss of ICP0 expression and thus the rapid recombination of our primary 3×-FLAG-RL2A mutant upon serial passaging. Furthermore, this may also explain some of the attenuation, which we observed for the mutant viruses with 3×-FLAG-tagged NTEs, namely for ICP27 and VP5. Accordingly, protein levels of the canonical ICP27 protein were dramatically reduced for the 3×-FLAG-tagged NTE-ICP27 virus (Supplementary Fig. 5). These observations highlight the need to carefully consider ectopic translation start site usages when manipulating herpesvirus genomes.

Transcription of all viral genes continuously increases throughout lytic infection with the exception of the transcript encoding ORF-O and ORF-P. These two partially overlapping ORFs are expressed antisense to the ICP34.5 (RL1) gene[46]. Consistent with the previous report, the respective transcript was already well detectable in 4sU-seq data at 1 h p.i. but transcriptional activity declined rapidly afterwards (Supplementary Fig. 7a). Nevertheless, translation of the respective transcripts remained detectable until late times of infection. Interestingly, the absence of a canonical start codon resulted in the hypothesis that ORF-O initiates from the same AUG start codon as ORF-P but then diverges within the first 35 codons due to a ribosomal frameshift. We did not observe any evidence for frameshifts in the HSV-1 translatome. While translation of ORF-O was rather weak, our data indicate that it rather initiates from a non-canonical ACG start codon 76 nt upstream of the ORF-P (Supplementary Fig. 7b).

Finally, we also aimed to validate the newly identified HSV-1 ORFs by whole-proteome mass spectrometry. We obtained triple-SILAC whole-proteome data from HSV-1 infected HFF at 0, 4, and 8 h p.i ($n = 3$). Furthermore, we performed whole-proteome mass spectrometry from primary lung fibroblasts infected with HSV-1 for 0, 4, 9, and 15 h. In total, this confirmed only 11 (6%) of the 186 ORFs and sORFs (Supplementary Data 7; excluding NTEs and NTTs) identified in this study. This rather small fraction is consistent with previous work on HCMV ORFs[2] and presumably reflects that the majority of viral sORFs are inherently unstable and rapidly degraded upon translation similar to their cellular counterparts. Nevertheless, they may play an important role in regulating translation of the viral ORFs encoded downstream of them. Furthermore, the identified large viral ORFs were expressed at substantially lower (~10×) levels than the previously identified viral protein-coding sequences (Supplementary Fig. 8).

**The complete nomenclature of HSV-1 gene products**. The large number of viral gene products required extension of the current nomenclature. We first compiled viral gene units comprising transcript isoforms, ORFs and regulatory entities, e.g., uORFs and uoORFs (Supplementary Data 3). A detailed description of the applied rules is provided in the Supplementary Methods. In brief, we fully maintained the current nomenclature for all ORFs in the reference annotation[1] and attributed each ORF to the most highly expressed transcript in its vicinity. Alternative transcript isoforms initiating within less than 500 nucleotides were labeled with "*" (extended) or "#" (truncated), e.g., UL13 mRNA *1. Finally, alternatively spliced transcripts were labeled with iso 1 and iso 2. Short ORFs (<100 aa), were named upstream ORF (uORF), upstream overlapping ORF (uoORF), internal ORF (iORF) and downstream ORF (dORF) in relation to the next neighboring large ORF. Any ORF for which no transcript could be identified to be responsible for its translation was labeled as orphan. An overview of the status of the various kinds of ORFs that we identified is shown in Fig. 6d. In accordance, any transcript,

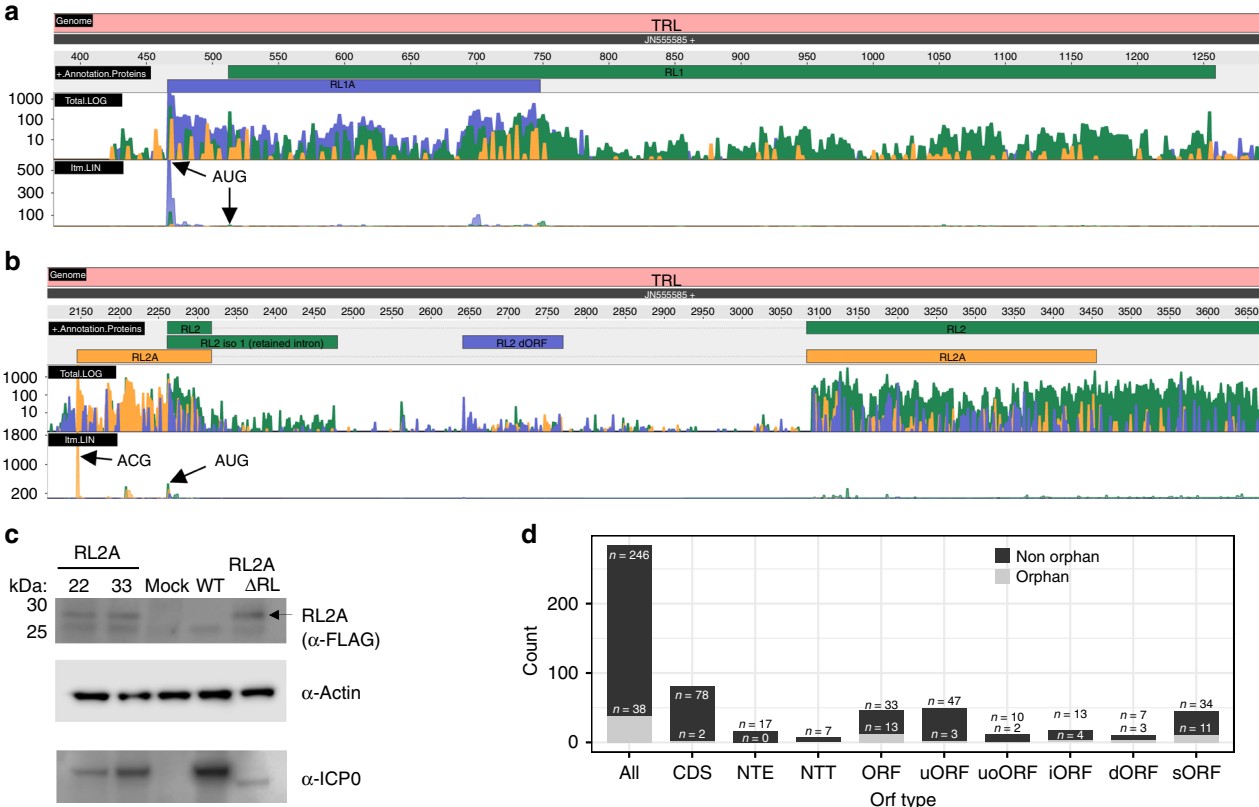

**Fig. 6 Evidence of additional protein-coding sequences.** Ribosome profiling data visualizing expression of the two viral open reading frames (ORFs), RL1A and RL2A (the three colors depicting the read count for each of the three possible frames, yellow = 1, blue = 2, green = 3), expressed from the **a** RL1 and **b** RL2 locus of the terminal repeats (TRL). Both standard ribosome profiling data in log scale (Total.log) as well as translation start site profiling data obtained using Lactimidomycin (linear scale, ltm.LIN) are shown. The three possible reading frames are colored in yellow (frame 1), blue (frame 2) and green (frame 3). Both ORFs are well expressed and validated by the strong peak of their respective Translation start site (TaSS) in the ltm-track (black arrows). While RL2A initiates from a non-canonical ACG start codon, the 93 aa RL1A protein initiates from an AUG start codon and was previously missed due to its length of <100 aa. **c** Validation of RL2A expression by Western blot. Primary human fibroblasts were infected with two viral clones (22, 33) with one RL segment expressing a 3×-FLAG-tagged RL2A (RL2A), mock, wild-type HSV-1 (WT; for 24 h) or 3×-FLAG-RL2A-ΔRL (=RL2A-ΔRL) for 24 h. Interestingly, insertion of the 3×-FLAG-tag resulted in a loss of ICP0 expression presumably due to the introduction of three out-of-frame AUG start codons (within each FLAG-tag) upstream of the ICP0 TaSS. This was most pronounced when the second repeat was deleted (RL2A-ΔRL). Actin served as house-keeping control. A representative experiment of two independent experiments is shown. **d** Distributions of all identified types of ORFs (CDS coding sequence, NTE N-terminal extended ORFs, NTT N-terminal truncated ORFs, uORF upstream ORF, uoORF upstream overlapping ORF, iORF internal ORF, dORF downstream ORF, sORF short ORF) of HSV-1 classified by ORFs and orphan ORFs. Source data are provided within the Source Data file.

which was not found to encode an ORF or sORF within its first 500 nt was also labeled as orphan. Interestingly, we identified 41 orphan transcripts (Supplementary Data 8), which showed predominantly nuclear localization indicating that they may represent viral nuclear long non-coding RNAs (lncRNAs). However, all of them were expressed at rather low levels. Accordingly, we were unable to validate five of them by Northern blots despite extensive efforts. We conclude that HSV-1 does not express any highly transcribed viral non-coding RNAs during lytic infection. We uploaded the fully reannotated HSV-1 genome information to the NCBI GenBank Third Party Annotation database (accession number BK012101).

## Discussion

In recent years, major advances in high-throughput experimental methodology have revealed that herpesvirus gene expression is surprisingly complex. While a number of studies in the last few years described hundreds of viral transcripts and ORFs, a systematic analysis, validation and integration into gene modules, which attribute individual ORFs and sORFs to specific transcripts

they are expressed from, was not attempted. Moreover, the lack of a standardized nomenclature has hampered functional studies on these viral gene products. Based on a wide spectrum of this and already published functional genomics data, we here provide a state-of-the-art, fully revised annotation of the HSV-1 genome.

Calling gene products based on big data poses the risk of false positives. As virus replication already initiates at 2 h p.i., first virus particles are released at 4 h p.i. and >80% of translational activity in the infected cells is viral at 8 h p.i.[11], we restricted our analysis to the first 8 h of infection to reduce the risk of detecting aberrant gene expression in cells with extensive cytopathic disruption of the transcriptional machinery. Integrative analysis of transcription start site (TiSS) data obtained by both second (cRNA-seq, dRNA-seq) and third (PacBio, MinION) generation sequencing approaches highlighted the necessity to validate viral TiSS by multiple means to exclude such experimental artifacts generated by the individual approaches. Similarly, the different transcription profiling approaches identified numerous putative splicing events. However, the vast majority of these only occurred at very low levels. We restricted our analysis to the first 8 h of lytic HSV-1 infection and only included splicing events observed by at least two approaches. While MinION

sequencing recently identified intergenic splice sites resulting in fusion proteins (e.g., between ICP0 and glycoprotein L)[14], the respective transcripts only arise very late in infection and their functional relevance remains unclear. We thus did not include them into our annotation. We conclude that splicing in the HSV-1 transcriptome was already well described by the previous reference annotation but rare splicing events may explain some of our orphan viral ORFs and sORFs.

In eukaryotic cells, RNA polymerase II (Pol II) may continue transcribing for thousands of nucleotides downstream of the PAS until transcription is terminated and Pol II is released from the chromatin[47]. With viral gene expression rapidly increasing throughout lytic infection, mRNA precursors with unprocessed 3′-ends that still extend beyond the canonical polyadenylation site are likely to be prevalent in the infected cells. Thus, unprocessed viral pre-mRNAs can easily be misinterpreted as mature viral transcripts. This presumably explains previous reports of near-complete transcription of herpesviral genomes during productive infections[2,48,49]. Analysis of cytoplasmic rather than total RNA provides a more accurate picture of the mature viral transcriptome. Consistent with previous reports, we confirmed that the major viral RNA export factor ICP27 was required for efficient export of all viral transcripts[32]. Interestingly, this also included all immediate early genes and spliced viral transcripts.

Ribosome profiling identified 134 sORFs expressed during lytic HSV-1 infection. The majority of these represent so called upstream open reading frames (uORFs). Interestingly, a relatively large fraction of transcript isoforms (~20%) encode their own uORF, which preferentially (54%) initiate from AUG start codons. Cellular uORFs constitute an important regulatory network governing gene expression at the level of translation by affecting translation initiation of the downstream ORF[50]. Reliable annotation of both viral transcripts and their respective uORFs will now enable functional studies on these cryptic viral gene products. sORF-encoded polypeptides are usually highly unstable and thus remain undetectable by whole-proteome mass spectrometry. Accordingly, we were only able to confirm about 5.5% of the ORFs identified by ribosome profiling on the peptide level. Interestingly, we could recently show that peptides derived from cellular sORFs are nevertheless efficiently incorporated into and presented by MHC-I molecules on the cell surface despite remaining virtually undetectable by whole proteome mass spectrometry[5]. Viral sORF-derived peptides thus may constitute a viral class of antigens that are efficiently presented by MHC-I but, due to their instability and extremely low abundance within the cell's proteome, represent poor substrates for cross-presentation and CD4-CD8 augmentation. Further studies are necessary to assess the role of HSV-1 sORFs in the regulation of viral protein expression, antiviral T cell control and evasion thereof.

Based on our revised annotation of 201 viral transcripts and 284 ORFs, we extended the existing nomenclature to include these all our novel viral gene products. This did not involve any renaming of previously described viral gene products. Our nomenclature thereby explains gene expression of the majority of viral ORFs in the context of different transcript isoforms, uORF and uoORFs. This will facilitate functional studies on the viral gene products as well as their transcriptional and translational regulation.

## Methods

**Cell culture, viruses, and infections.** Human foreskin fibroblasts (HFF, #86031405, purchased from ECACC), 293T, Vero 2–2 (Smith, Hardwicke, & Sandri-Goldin, 1992) BHK-21, and BHK-21 dox-UL19 (described below) cell lines were cultured in flasks containing Dulbecco's Modified Eagle Medium (DMEM), high glucose, pyruvate (ThermoFisher #41966052) supplemented with 1x MEM Non-Essential Amino Acids (ThermoFisher #11140050), 1 mM additional sodium pyruvate (ThermoFisher #11360070), 10% (v/v) Fetal Bovine Serum (FBS, Biochrom #S 0115), 200 IU/mL penicillin (pen) and 200 μg/mL streptomycin (strep).

WI-38 fibroblasts were cultured in minimal essential media (MEM) supplemented with 10% FBS, 100 IU/mL penicillin, and 200 μg/mL strep. All cells were incubated at 37 °C in a 5% (v/v) CO$_2$-enriched incubator.

HFF were utilized from passage 11–17 for all high-throughput experiments. Virus stocks were produced on baby hamster kidney (BHK) cells except for the viruses described below. Stocks of the ICP27-null mutant (strain KOS)[51] were produced on complementing Vero 2–2 cells[52]. HFF were infected for 15 min at 37 °C about 24 h after the last split using a multiplicity of infection (MOI) of 10. Subsequently, the inoculum was removed and fresh media was applied to the cells.

To reconstitute the 3×-FLAG-tagged UL19 NTE, BHK-21 dox-UL19 cells with doxycycline-inducible expression of UL19 were generated by cloning the HSV-1 Syn17 + UL19 coding sequence using primers described in Supplementary Data 9 into the SalI and NheI sites of pTH3, a derivative of pCW57.1 with a custom multiple cloning site in lieu of the gateway cloning site and the addition of the TRE tight promoter from pTRE-Tight. Lentiviral vectors were generated by cotransfection of this construct with psPAX2 and pCMV-VSV-G into 293T cells. Lentivirus-containing supernatants were sterile-filtered with Minisart® NML 0.45 μm cellulose acetate filters (Sartorius #17598) and added to BHK-21 cells. Polyclonal populations were selected 48 h post-transduction and maintained in 1 μg/mL puromycin.

**Viral mutagenesis and reconstitution.** All viral mutants were generated via en passant mutagenesis[53] using *Escherichia coli* strain GS1783 with the bacterial artificial chromosome (BAC) HSV1(17+)-LoxCheVP26[54] expressing a fusion protein of mCherry on the N-terminus of the *UL35* gene product (VP26). Full primer and construct sequences can be found in Supplementary Data 9. BAC DNA was purified using the NucleoBond BAC 100 kit (Macherey-Nagel #740579) and transfected for virus reconstitution into BHK-21 cells with Lipofectamine 3000 (ThermoFisher #L3000-075). HSV-1 expressing the 3×-FLAG-tagged N-terminal extension of UL54 was reconstituted and titrated on Vero 2–2 cells[52]. The virus expressing the tagged N-terminal extension of UL19 was generated in BHK-21 dox-UL19 cells. BHK-21 dox-UL19 cells were plated the day before in media containing 1 μg/mL doxycycline (Sigma #D3072), which was maintained throughout virus generation. 3×-FLAG-tagged RL2A BAC-derived viruses were constructed by insertion of the tag with a kanamycin cassette into one genomic repeat followed by replacement of the second repeat (region upstream of RL1 through the second exon of RL2) with the ampicillin resistance gene from pcDNA3. The kanamycin cassette was removed thereafter by traceless mutagenesis.

Virus produced by transfected cells was expanded on minimally five T175 flasks of the corresponding cell type. Virus-containing supernatants were harvested upon >90% cytopathic effect and centrifugation at 8000 RCF at 4 °C for 10 min to pellet cells. Cell pellets were snap-frozen in liquid nitrogen and thawed at 37 °C three times to free cell-associated virus. Cellular debris was pelleted at 10,000 RCF, 4 °C for 10 min and supernatant combined with the supernatant in the previous step. Virions were pelleted by centrifugation at 19,000 RCF for two hours at 4 °C, resuspended in phosphate-buffed saline (PBS), and pelleted once more over a 20% (w/v) sucrose cushion in PBS 16,000 RPM for 2 h at 4 °C in a SW 28 swinging-bucket rotor (Beckman). Virus pellets were resuspended in PBS, snap-frozen in liquid nitrogen, stored at −80 °C, and titrated by plaque assay. Infections were carried out in serum-free DMEM containing penicillin and streptomycin for 1 h at 37 °C. The time at which inoculum was replaced with growth media was marked as the 0 h timepoint.

**Western blot.** Samples were harvested at the indicated timepoints by removal of growth media and direct lysis in 2x Laemmli buffer containing 5% (v/v) β-mercaptoethanol. Samples were sonicated and heated for 5 min at 95 °C before loading onto a Novex™ WedgeWell™ 4–20% Tris-Glycine Gel (ThermoFisher #XP04200BOX). Proteins were transferred to polyvinylidene difluoride (PVDF) membranes, blocked for 1 h at room temperature in 1xPBST containing 5% (w/v) milk (Carl Roth T145.3), and probed using α-FLAG M2 (Sigma #F1804) overnight at 4 °C at a 1:1000 at dilution and α-mouse IgG (whole molecule)-peroxidase (Sigma #9044) for 1 h. β-actin was probed using α-β-actin C4 antibody (Santa Cruz Biotechnology #sc-47778) at a 1:1000 dilution for 1 h, followed by IRDye® 800CW goat α-mouse IgG (LI-COR #926-32210) at 1:5000 or α-mouse IgG (whole molecule)-peroxidase (Sigma #9044) for 1 h. ICP0 and ICP27 were probed using α-ICP0 clone 5H7 (Santa Cruz Biotechnology #sc-56985) or α-ICP27 clone H1113 (Abcam #ab53480), respectively, at a 1:1000 dilution for 1 h, followed by IRDye® 680RD goat α-mouse IgG (LI-COR #926-68070) at 1:5000 or α-mouse IgG (whole molecule)-peroxidase (Sigma #9044) for 1 h. Samples were washed with 1xPBST and blocked before addition of each antibody in the milk/PBST buffer. Blots were visualized with a LI-COR Odyssey® FC Imaging System.

**Immunofluorescence.** 10$^5$ HFF cells were plated on glass coverslips in 12-well dishes 24 h prior to infection. At 8 h post-infection, cells were fixed in 4% formaldehyde in PBS for 1 h at room temperature, washed three times in PBS and stored at 4 °C overnight in PBS. Cells were incubated in permeabilization buffer (10% FBS, 0.25 M glycine, 0.2% Triton X-100, 1xPBS) for 1 h at room temperature before incubating them in blocking buffer (10% FBS, 0.25 M glycine, 1xPBS) for 1 h at room temperature. Anti-FLAG antibody (GenScript #A00187) was incubated in

10% FBS and 1xPBS for 1 h at 37 °C at a concentration of 1 μg/mL. The secondary anti-mouse IgG, Alexa Fluor 488 (ThermoFisher #A11017) was incubated in 10% FBS in 1xPBS for 1 h at room temperature with 0.5 μg/mL 4',6-diamidino-2-phenylindole (DAPI). All steps were followed by three 5-min washes in PBS except for after the primary antibody, which was washed with 1xPBS and 0.05% Tween-20. Coverslips were washed in water before mounting them in medium containing Mowiol 4-88 and 2.5% (w/v) 1,4-diazabicyclo[2.2.2]octane (DABCO).

**Transcription start site (TiSS) profiling**. Total cellular RNA was isolated using Trizol (Invitrogen) following the manufacturer's instructions. RNA was resuspended in water and stored at −80 °C until use. TiSS profiling dataset using cRNA-seq utilizes a similar approach as employed for decoding HCMV[2]. Following rRNA depletion and extensive chemical RNA fragmentation, 50–80 nt RNA fragments are recovered by gel extraction. Library preparation is performed by 3′-adaptor ligation and circularization. This inherently enriches for transcript 5′-ends by 20- to 30-fold. A detailed protocol is included in the Supplementary Methods. Of note, our cRNA-seq library preparation protocol introduces a 2 + 3 nt unique molecular identifier (UMI), which facilitates the subsequent removal of PCR duplicates from sequencing libraries.

TiSS profiling dataset using dRNA-seq was prepared according to the published protocol[12] with some modifications by the Core Unit Systems Medicine (Würzburg). In brief, for each sample 3 μg of DNase-digested RNA was treated with T4 Polynucleotide Kinase (NEB) for 1 h at 37 °C. RNA was purified with Oligo Clean & Concentrator columns (Zymo) and each sample was split into an Xrn1 (+Xrn1) and a mock (−Xrn1) sample. The samples were treated with 1.5 U Xrn1 (NEB; +Xrn1) or water (−Xrn1) for 1 h at 37 °C. Digest efficiency was checked on a 2100 Bioanalyzer (Agilent) and 5′ caps were removed by incubation with 20 U of RppH (NEB) for 1 h at 37 °C. Afterwards, RNA was purified and eluted in 7 μL and 6 μL were used as input material for the NEBNext® Multiplex Small RNA Library Prep Set for Illumina®. Library preparation was performed according to the manufacturer's instruction with the following modifications: 3′ adapter, SR RT primer and 5′ adapter were diluted 1:2, 13 cycles of PCR were performed with 30 sec of elongation time, and no size selection was performed at the end of library preparation. Concentrations of libraries were determined using the Qubit 3.0 (Thermo Scientific) and their fragment sizes were determined using the Bioanalyzer. Libraries were pooled equimolar. Sequencing of 75 bp single-end reads was performed on a NextSeq 500 (Illumina) at the Cambridge Genomic Services (cRNA-seq) and the Core Unit Systems Medicine in Würzburg (dRNA-seq). To validate TiSS identified by cRNA-seq, dRNA-seq, PacBio or MinION (no reads were analyzed, only the called transcripts were used for PacBio and MinION), total RNA-seq and 4sU-seq data that were previously published[11] were reanalyzed (see below).

**RNA-seq of subcellular RNA fractions**. Subcellular RNA fractions (cytoplasmic, nucleoplasmic and chromatin-associated RNA) were prepared by combining two previously published protocols[55,56]. Data from uninfected and wild-type HSV-1 infected cells were recently used recently[15]. Infection with the ICP27-null mutant was performed in the same experiment. As for wild-type HSV-1 infection, total cellular RNA was isolated using Trizol at 8 h p.i. Fractionation efficiencies were confirmed on the RNA-seq data by comparing expression values of known nuclear and cytoplasmic RNAs as well as intron contributions (see Supplementary Fig. 9)[15]. Sequencing libraries were prepared using the TruSeq Stranded Total RNA kit (Illumina) following rRNA depletion using Ribo-zero. Sequencing of 75 bp paired-end reads was performed on a NextSeq 500 (Illumina) at the Cambridge Genomic Services and the Core Unit Systems Medicine (Würzburg).

**Ribosome profiling**. The ribosome profiling time-course data (lysis in presence of cycloheximide) have already been published[11]. Additionally, so far unpublished data we generated include translation start site (TaSS) profiling performed by culturing cells in medium containing either Harringtonine (2 μg/ml) or Lactimidomycin (50 μM) for 30 min prior to harvest. Harringtonine samples were obtained for 2 h and 8 h p.i., Lactimidomycin was employed for mock, 4 and 8 h p.i. Two replicates of each condition were analyzed. As described for cRNA-seq, the library preparation protocol introduces a 2 + 3 nt unique molecular identifier (UMI), which facilitates the subsequent removal of PCR duplicates from sequencing libraries. All libraries were sequenced on a HiSeq 2000 at the Beijing Genomics Institute in Hong Kong.

**Proteomic analysis**. WI-38 lung fibroblasts grown in SILAC medium were infected at MOI 10 and harvested after 4, 9, and 15 h. After cell lysis, protein concentration was determined by Bradford assay and 200 μg of each sample were mixed with equal amount protein extracted from untreated cells grown in SILAC light medium. Additional sample preparation steps are elaborated in the Supplementary Methods. Primary human foreskin fibroblasts were grown for five passages in DMEM lacking lysine and arginine (Thermo Scientific) supplemented with 10% dialysed FCS (Gibco), 100 units/mL penicillin and 0.1 mg/mL streptomycin, 280 mg/L proline (Sigma) and light (K0, R0; Sigma), medium (K4, R6; Cambridge Isotope Laboratories) or heavy (K8, R10; Cambridge Isotope Laboratories) 13 C/15N-containing lysine (K) and arginine (R) at 50 mg/L. Pre-labeled cells were

infected with HSV-1 at a multiplicity of infection (MOI) of 10 for 4 or 8 h, and uninfected cells were included as a control. The experiment was conducted in triplicate (biological replicates), with a 3-way SILAC label swap. The full description of sample processing is continued in the Supplementary Methods file.

**Data analysis, statistics, and reproducibility**. Western blotting and immunofluorescence images are representative of at least two independent biological replicates. Random and sample barcodes in cRNA-seq and ribosome profiling data were analyzed by trimming the sample and UMI barcodes and 3' adapters from the reads using our in-house computational genomics framework gedi (available at https://github.com/erhard-lab/gedi). Barcodes introduced by the reverse transcription primers included three random bases (UMI part 1) followed by four bases of sample-specific barcode followed by two random bases (UMI part 2). Reads were mapped using bowtie 1.0 against the human genome (hg19), the human transcriptome (Ensembl 75) and HSV-1 (JN555585). The HSV-1 genome consists of two components (L and S) that are both flanked by long repeats. To mitigate the effect of multi-mapping reads, we masked the terminal repeats by NNN. The three mappings were merged and only the alignments for a read with minimal number of mismatches were retained. Reads were assigned to their specific samples based on the sample barcode. Barcodes not matching any sample-specific sequence were removed. PCR duplicates of reads mapped to the same genomic location were identified by counting UMIs. If two observed UMI differed by only a single base, one likely is due to a sequencing error. Thus, we discarded one of the two in such cases. If the reads at this location mapped to k locations (i.e., multi-mapping reads for k > 1), a fractional UMI count of 1/k was used (see Supplementary Table 1 for read yields)[11]. Finally, all read mappings in the repeats were copied into the previously masked regions.

dRNA-seq, 4sU-seq, total RNA-seq and RNA-seq data of subcellular fractions were processed similar to cRNA-seq and ribosome profiling data with the exception of STAR (v.2.5.3a) being used to map the reads and PCR duplicates were not collapsed as no random barcodes were used (see Supplementary Table 1 for read yields).

All libraries were prepared with strand-sensitive protocols. Consequently, all reads were mapped only to their respective strand. Further, reads were weighted by the number of different locations they map to (i.e., if a read mapped to three locations, its weight was 1/3).

Our dRNA-seq and cRNA-seq TiSS profiling data were analyzed with our TiSS analysis pipeline iTiSS (integrative Transcriptional Start Site caller, see Supplementary Methods), which identifies potential TiSS at single-nucleotide resolution. Briefly, dRNA-seq and cRNA-seq data were searched for positions showing strong accumulation of reads 5' ends compared to their surroundings using a sliding-window approach. Transcriptional activity was defined by having a stronger transcriptional activity downstream of a potential TiSS than upstream identified using Fisher's exact test. Selective induction or repression (altered expression kinetics) of the respective transcripts during the course of HSV-1 infection was defined by observing a significant change of transcriptional activity surrounding a potential TiSS during the course of infection using a likelihood ratio test based on the dirichlet distribution.

Each position in the viral genome could therefore score up to four points by iTiSS. However, we extended the total number of criteria to 9 by including additional evidence for the presence of a TiSS of other datasets or groups. The final list of criteria are as follows (criteria i–iv are checked by iTiSS. For a more detailed description see Supplementary Methods.):

(i) Significant accumulation of the 5′-end of reads in both replicates of the dRNA-seq dataset at a single position of the HSV-1 genome.

(ii) Significant accumulation of the 5′-end of reads in both replicates of the cRNA-seq dataset at a single position of the HSV-1 genome.

(iii) Stronger transcriptional activity downstream than upstream of the potential TiSS in both cRNA-seq replicates.

(iv) Significant temporal changes in TiSS read levels during the course of infection in both cRNA-seq replicates.

(v) A TiSS called in the MinION dataset with a maximum distance of 20 nt downstream.

For this purpose, the transcripts provided in the Supplementary Table of Depledge et al.[14] were taken.

(vi) A TiSS called in the PacBio dataset in close proximity (±5 nt).

For this purpose, we manually corrected the GFF-file provided alongside the GEO-submission for the PacBio data, which was inconsistent with the transcripts reported in the paper.

(xii) Stronger transcriptional activity downstream than upstream of the potential TiSS in both 4sU-seq replicates.

(viii) Significant temporal changes in TiSS read levels during the course of infection in both 4sU-seq replicates.

(ix) The presence of an ORF at most 250 bp downstream, which was not yet explained by another transcript.

Afterwards, potential TiSS within a ±5 bp window were combined into a single TiSS. Consequently, a TiSS is defined by a single-nucleotide position including a ±5 bp window. The fidelity of this definition can be appreciated by the strong enrichment of the Inr motif even for the most weakly expressed viral transcripts.

The reported enrichment factors for dRNA-seq and cRNA-seq were calculated based on predicted TiSS in human rather than HSV-1. This was done to prevent undesired biases due to read-in caused by the extraordinary high number of overlapping transcripts in HSV-1. The predicted TiSS were ordered based on the number of reads starting at their respective positions. The median was then calculated over the 50 strongest and 10 strongest expressed TiSS for cRNA-seq and dRNA-seq, respectively.

Significance of the correlation between the presence of a TATA-box-like motif and the transcription strength of TiSS was calculated using Fisher's exact test. Here, the TiSS were ordered by their numbers of reads starting and sorted into three equally sized bins. For the bin containing the strongest TiSS as well as the bin containing the weakest TiSS, the number of all nucleotides between position $-30$ and $-25$ relative to the TiSS were summed up. For the parameters of the Fisher's exact test, the following sums were used $a = T + A$(strongest bin), $b = C + G$ (strongest bin), $c = T + A$(weakest bin) and $d = C + G$(weakest bin).

We used our in-house tool PRICE[5] version 1.0.1 to call ORFs separately for the two replicates of ribosome profiling data but pooling all samples from each replicate.

RNA-seq data were mapped using STAR[57] version 2.5.3a using a combined reference index derived from Ensembl 90 and our final HSV-1 annotation.

We analyzed mass spectrometry data using MaxQuant[58] version 1.6.5.0. Spectra were matched against a combined database of proteins from Ensembl (version 75), and all ORFs identified by ribosome profiling. We used carbamidomethylation as fixed and acetylation (N-terminal) and oxidation at methionine as variable modifications. Peptides were filtered for 1% FDR using the target-decoy approach by MaxQuant.

The export indices of chromatin-associated RNA and cytoplasmic RNA were derived by computing their fold changes between the wild-type and the null mutant for ICP27 using the lfc R-package[59]. Data handling and visualization was done using R including the ComplexHeatmap[60], circlize[61], ggseqlogo, ggplot2, reshape2, plyr, scales, ggforce, ggrepel, and the gridExtra packages.

To reveal the potential function or functional motifs of predicted protein sequences, we used sequence comparison, domain composition, structure prediction and motif searches[62–64]. Sequence comparisons exploited Blast searches iteratively[65] and identified catalytic as well as regulatory domains including predictions by the conserved domain database[66]. Predicted domain composition was verified using domain databank tools SMART[67] and Prodom[68]. Motif searches exploited Prosite regular expressions and profiles and the integrative protein signature database[69]. As independent tests for resulting function assignments structure annotation for protein domains was done using AnDOM software[70] as well as homology predictions by SwissModel[71]. Gene context methods were applied for unclear sequences related to non-viral sequences (STRING database[72]). In addition, Clusters of Orthologous Groups using the latest version (5.0) of the eggNOG tool with its 2502 virus strains provided independent annotation input[73].

**Reporting summary.** Further information on research design is available in the Nature Research Reporting Summary linked to this article.

## Data availability

All datasets generated and/or analyzed during the current study have been deposited in the Gene Expression Omnibus (GEO) database with accession codes GSE128324 (Translation start site profiling, transcription start site profiling), GSE59717 (4sU-seq and total RNA-seq), GSE60040 (ribosome profiling), and GSE128880 (cytoplasmic, nucleoplasmic and chromatin-associated RNA). We only used the transcripts predicted in the PacBio and MinION dataset, respectively. However, the raw data can be found at the GEO database with accession code GSE97785 (PacBio)[13], and from the European Nucleotide Archive (ENA) with the study accession code PRJEB27861 (MinION)[14]. The mass spectrometry proteomics data have been deposited to the ProteomeXchange Consortium via the PRIDE partner repository with the dataset identifiers PXD013010 and PXD013407. The revised annotation can be accessed at NCBI GenBank Third Party Annotation database under the accession number BK012101. The source data underlying Figs. 1, 2b–d,f, 3a, 4a–d, 5a–g, 6c, and Supplementary Fig. 5a–d are provided as a Source Data file.

## Code availability

Scripts used to create the Figures and Supplementary Data and Tables and analyze the omics data are available at Zenodo[74].

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

## Acknowledgements

This work was supported by the European Research Council (ERC-2016-CoG 721016 – HERPES to L.D.), the MRC (CSF G1002523 to L.D. and MR/P008801/1 to N.J.M.), the Wellcome Trust (PRF 210688/Z/18/Z to P.J.L.), NHSBT (WP11-05 to L.D. and WPA15-02 to N.J.M.), DFG (1275/6-1 to L.D., GR950/16-1 to F.G., LA2941/4-1 to M.L., SFB1123/Z2 to R.Z. and FR2938/7-1 to C.C.F.), the NIHR Cambridge BRC, a Wellcome Trust Strategic Award to CIMR and the IZFK at the University of Würzburg (project Z-6). A.W.W. was the recipient of a generous grant from the Alexander von Humboldt Foundation and the German Federal Foreign Office. This publication was supported by the Open Access Publication Fund of the University of Wuerzburg.

## Author contributions

A.W.W., T.H., E.W., B.P., A.L.H., L.Dj., M.G., K.D., J.M., A.J.R., R.A., N.J.M., F.W.H.K, G.M., C.B., S.K., and F.G. performed the experiments. A.W.W., C.S.J., F.E., and L.D. designed the experiments, analyzed the data and wrote the paper. C.S.J., C.C.F., and F.E. performed the computational analyses. R.Z. supervised development of the computational methods (for ribosome profiling analysis). S.K., M.L., and P.J.L. supervised the mass spectrometry analyses and C. L. and T.D. contributed to motif analysis for the viral ORFs.

## Competing interests

The authors declare no competing interests.
