## [Peer Review File · Nature Communications]

Reviewers' Comments:

Reviewer #1:

Remarks to the Author:

The authors have queried a collection of NGS datasets to expand and refine our understanding of the transcriptomic and proteomic output of herpes simplex virus during infection of primary human fibroblasts. Some of the datasets are new, others have been published previously, either by the authors themselves or by other groups. As one might expect from recent studies of related viruses, applying new technology and integrating both long-read and short read sequencing, yields a more complex landscape of transcript isoforms that recognized by the now dated HSV-1 reference annotation. In addition to the transcript maps, other features are cataloged including transcription start sites (TiSS) and accompanying core promoter elements, RNA splicing events, regulated mRNA export into the cytoplasm, polyA signal readthrough and so on. There's lots of interesting information buried in the data but in general, the biological significance of the new features is not well developed.

Arguably, most herpesvirologists will be keenly interested in the expanded number of ORFs determined by ribosome profiling (Ribo-seq) across five timepoints. All of the canonical ORFs are confirmed but some 46 new ORFs (defined as >100 codons in length) are reported, together with 134 small ORFs (defined as <100 codons). Studies in HCMV and KSHV have highlighted the extensive use of non-AUG codons by herpesviruses and this holds true for HSV-1: of the 284 ORFs delineated by the authors, 21% begin with a non-AUG codon. The discovery of N-terminal extensions to a number of well-studied proteins (16 out of the 80 or so canonical proteins) is especially interesting. These include extensively researched proteins such as the multifunctional regulator ICP27 and the AKT mimic Us3. The authenticity of six of these was tested by making recombinant viruses in which three copies of the FLAG epitope were placed within the extension or downstream of the canonical ATG. In a couple of cases the tag rendered the virus non-viable but others were successful. Both the viral kinase Us3 and the viral dUTPase (UL50) showed different subcellular localization between the extended form and the canonical protein. The consequences with respect to the known functions and substrates of these enzymes were not tested further.

Similarly, the existence of a second ORF (here named RL2A) overlapping the well-studied RL2 CDS encoding ICP0 was tested by the triple FLAG insertion approach. However, the tagged virus is compromised somehow and disappears shortly after reconstitution. This is a potentially very interesting observation but again needs to be fleshed out better. Does provision of the untagged RL2A polypeptide in trans prevent loss of the FLAG-tagged mutant? Is viability compromised by insertion of the tag into RL2A (implying the functional polypeptide is critical) or through some effect on the expression of ICP0? The arrangement of the ORFs resembles an upstream overlapping ORF (uoORF) and like other examples might function to regulate translation efficiency of the downstream RL2. The fact that RL2A uses a non-canonical initiation codon (ACG) may be relevant if there is some regulatory connection. Some of these possibilities can be addressed using the isolated transcription unit, independent of the whole virus.

Revising an existing genome annotation – especially one as partisan and entrenched as that of HSV-1 – is always difficult. The authors are to be commended for laying out their logic carefully as a supplementary document. Some transcript isoforms are put aside due to very low abundance. The challenge of validation is a recurring issue in studies advocating expanded viral (or cellular) proteomes and the use of Ribo-seq seems a wise choice. By triple-SILAC whole proteome mass-spec, the authors were able to confirm fewer than 6% of the predicted ORFs. Whether this reflects instability, low expression or both remains a matter of speculation.

Overall the manuscript covers a lot of ground and touches only superficially on many interesting and

potentially important observations. Of course, there is value to making the data available to broader community in a timely fashion but as the manuscript stands, too much is left requiring further validation or some attempt to explore the biological relevance.

SPECIFIC COMMENTS

The new nomenclature uses various symbols to denote extended or truncated transcripts. Will these be clear enough on figures and other maps to be maintained?

Pg 6 line 208 "Expression of both a 135 kDa (full-length) and 90kDa (NTT) protein has been shown for the commercial ICP8 antibody 11E2 (Santa Cruz, see product website)." There's a sizeable literature on ICP8 and yet this truncated isoform has not been commented on before. Commercial antibodies are less specific than advertised and it seems odd to cite a non-peer reviewed source. Can this not be tested directly? According to the website image, HSV-1 (MacIntyre strain) infected African Green monkey kidney do not produce the shorter species seen the HSV-1 (17 syn + strain) infected baby hamster kidney cells. Are there clues in the genomic sequence (GenBank accession no. KM222720) as to why this might be?

For the most part, the authors limit their analysis to the first 8 hours of infection but the rationale for this and its relationship to major landmarks in the replication cycle (onset of DNA synthesis, appearance of new infectious particles etc) is not laid out.

Unfortunately, I was unable to run the Mac OS version of the genome browser and thus could not assess its utility as a resource.

Some preprints are cited that are now published. These should be updated and consolidated.

Reviewer #2:

Remarks to the Author:

The paper of Whisnant et al. titled 'Integrative functional genomics decodes herpes simplex virus 1' reports a comprehensive analysis of several sets of high-throughput sequencing data of HSV-1 infected cells, leading to comprehensive annotation of HSV-1 transcripts and corresponding ORFs. The main novelty of the paper is the heroic effort to combine multiple types of sequencing data and to integrate it to an accessible resource that can help follow-up genetic studies.

Below are several comments that can help to improve clarity:

1. Data-sets used in the paper: Some of the analyzed datasets were created by new experiment, while others were previously published. Although the novelty of each dataset is stated in the legend of figure 1, throughout the text the distinction between novel and reanalyzed data is not stated clearly enough (in the overview rows 83-90, in the data availability rows 619-627). Additionally, the MinION data is attributed to two separate sources at different places in the text (ref 12 in line 86 and in figure 1 legend line 840, and ref 14 in line 108). State more clearly how the long-read data was used. Currently the writers state regarding the PacBio data that a modified GFF file was used (line 576) with no details for the MinION data. Specifically, Depledge et al. paper (ref 14 and 29) published in February 2019 identified transcription start sites and termination sites using MiniION and a very thorough analysis. This publication includes an organized table of all identified transcripts and the ORFs downstream to them. The authors should directly address these previously published results and compare them to the new annotations presented in this manuscript. If they are overlapping they

should acknowledge that , if not they should clarify the differences.

Since differential PolyA and read-through is reported, it is important to detail how full transcripts (TiSS to polyA) were annotated (meaning, how it was decided which PAS belongs to which TiSS). The same is true for any alternative splicing events, how were these compiled in to the annotations?

2. References and previous work:

As indicated above there are two published MinION datasets it is not clear which one was used and how.

The only papers cited for sORF function are for uORFs (line 76), some more can be cited for other functional sORFs.

Some papers appear twice in the reference: ref 9 and 55, ref 14 and 29

In line 169 the authors cite ref 30 seemingly out of context

3. Splicing:

- the authors show identification of splice junctions but do not specify with which tools where used to detect them

- figure s2 is hard to understand. Are these reads spanning splice junctions? What does upstream/downstream mean in this context?

4. data presented in figures

Fig 2:

A- The y axis should be labeled.

B-C it is unclear why the MinION and PACbio identified TiSS are shown on separate diagrams. Also there are two studies using MinION. Which data was used here?

D- the criteria of TiSS annotation. It is unclear why the presence of an ORF downstream of an initiation should be a requirement for the annotation of a transcript.

Fig 3:

A- The difference between the coverage drop after the PAS between the cytoplasm and the other fractions is not very convincing (figure 3a). Some statistic should be added and a gene example to strengthen the point.

B- Since multiple types of RNAseq data were used, it should be specified which one is shown in the genome browser figure 3b. The y axis should be labeled.

Fig 5:

This figure includes a novel interesting finding, that extension of viral proteins can support different localization. However there are several issues that need to be addressed here

A-E

- Requires explanation as it is strange that the NTE appears only when it is tagged upstream of the main AUG, and not when the main AUG is tagged, since the location of the tag should not affect the expression level. Wouldn't it made more sense to have one tag at the C-terminal that allow assessment of the relative expression by seperating the bands.

Only in UL54 two bands are seen when the main AUG is tagged but here there is very different intensity compared to the tagging of the NTE, again why? Altogether these raise the concern the expression of the NTE is neglect-able compared to the main AUG. how does this goes along with the measurements of above 10%?

F-G

The immunofluorescence of the NTE tagged genes is very weak compared to the AUG tagged ones. This is probably due to expression level differences, however, the weak signal (especially that of US3 NTE) makes it hard to distinguish from background noise. It will be better show it in comparison to background staining (no primary Ab).

In addition the preparation of only 2 out of 5 viral mutants are detailed in the methods section. The gene names should be consistent (UL54 in the figure and ICP27 in the main text, etc.)

Fig 6:

The different colored frames should be explained clearly (there are 2 types of green). The y axis should be labeled.

C- The expected and observed size should be clearer.

5. Genome Browser:

The genome browser provided for viewing the data is easy to install and navigate.

Some notes: the searching for terms only works when they are at the beginning of the gene names, and the double clicking on genetic element does not work. In addition, easy viewing of data will benefit from adding the option to minimize the size of tracks (such as squished in IGV) or selecting partial tracks out of the main ones (e.g. translation).

Minor comments:

Line 81 "Functional genomics" is used out of context since the annotation here is not of function

Line 141 add "the" before core

Line 186 selective should be selected

Line 271 delete only

Adding arrows to indicate the strand of current gene will facilitate the understanding of genome browser figures (especially the ones in the supplementary).

Reviewer #3:

Remarks to the Author:

The authors of this work have an interesting and 'hot' topic and the paper reads well. However, the actual details and substance are very difficult to follow. It has taken me a long time and after a considerable struggle I am sure that there is no way that I can reproduce their work. Further they have a whole new software and the description of the metrics the software uses are cryptic. This is a black box. It is clear the authors have done a lot of work, that is why I initially asked for a full set of methods. The response was somewhat helpful but not nearly detailed enough. I have identified some examples to help guide the authors, but this is far from an exhaustive list. This is so poorly described that it needs a complete rewrite. Someone needs to go slowly step by step and describe 1) how many samples and where they came from 2) the qc metrics 3) the detailed analysis protocols so that these can be reproduced 4) the rationale for each step 5) the detailed raw results for each rep should be provided.

I am not sure if the MASS Spec Data are not deposited.

Characterization of transcriptome, 155 TiSS page 4 line 101, 204 TiSS page 4 line 114. So what is 204? the union of TiSS and dRNA? This type of ambiguity is found throughout the paper.

Line 124 'scores for differences' HOW? The new supplement was very confusing... for example "If the read start count for a positing exceeded an interquartile range of 5, it was considered a potential TiSS and marked as such. If the same position was marked as a potential TiSS in both replicates, the position's TiSS-score was incremented by one." What does this mean? That the difference in the 75 and 25 quartiles should be bigger than 5 reads? And then "If the same position was marked as a potential TiSS in both replicates, the position's TiSS-score was incremented by one." So what was the start score?

"This was done using Fisher's exact test by defining a threshold (mean read start count downstream) and counting the number of positions that have a lower or higher read start count for the upstream and downstream window, respectively. If there was a significant difference ($p\text{-value} < 0.01$), the position was marked as a potential TiSS." So Fisher's exact test is done on a contingency table. I can't figure out 1) what is being counted in the table and 2) how many millions of tests they have done. For sure a pvalue of 0.01 is liberal.

I can't figure out how many replicates there are of anything except the TiSS which in one line says 2 but the others are not clear- you need to go to the data deposit and there the numbers are disappointing.... what remains unclear is whether these are the same biological samples shared between techniques or are they independent?

The data analysis is completely cryptic. Many of the sentences do not make sense.

Comments to the reviewers

Reviewer #1 (Remarks to the Author):

The authors have queried a collection of NGS datasets to expand and refine our understanding of the transcriptomic and proteomic output of herpes simplex virus during infection of primary human fibroblasts. Some of the datasets are new, others have been published previously, either by the authors themselves or by other groups. As one might expect from recent studies of related viruses, applying new technology and integrating both long-read and short read sequencing, yields a more complex landscape of transcript isoforms that recognized by the now dated HSV-1 reference annotation. In addition to the transcript maps, other features are cataloged including transcription start sites (TiSS) and accompanying core promoter elements, RNA splicing events, regulated mRNA export into the cytoplasm, polyA signal readthrough and so on. There's lots of interesting information buried in the data but in general, the biological significance of the new features is not well developed. Arguably, most herpesvirologists will be keenly interested in the expanded number of ORFs determined by ribosome profiling (Ribo-seq) across five timepoints.

We would like to thank this reviewer for the positive and careful evaluation of our work. We feel that this paper closes a major gap in the field, namely a fully revised, state-of-the-art annotation of the genome of an important human pathogen.

All of the canonical ORFs are confirmed but some 46 new ORFs (defined as >100 codons in length) are reported, together with 134 small ORFs (defined as <100 codons). Studies in HCMV and KSHV have highlighted the extensive use of non-AUG codons by herpesviruses and this holds true for HSV-1: of the 284 ORFs delineated by the authors, 21% begin with a non-AUG codon. The discovery of N-terminal extensions to a number of well-studied proteins (16 out of the 80 or so canonical proteins) is especially interesting. These include extensively researched proteins such as the multifunctional regulator ICP27 and the AKT mimic Us3. The authenticity of six of these was tested by making recombinant viruses in which three copies of the FLAG epitope were placed within the extension or downstream of the canonical ATG. In a couple of cases the tag rendered the virus non-viable but others were successful. Both the viral kinase Us3 and the viral dUTPase (UL50) showed different subcellular localization between the extended form and the canonical protein. The consequences with respect to the known functions and substrates of these enzymes were not tested further. Similarly, the existence of a second ORF (here named RL2A) overlapping the well-studied RL2 CDS encoding ICP0 was tested by the triple FLAG insertion approach. However, the tagged virus is compromised somehow and disappears shortly after reconstitution. This is a potentially very interesting observation but again needs to be fleshed out better. Does provision of the untagged RL2A polypeptide in trans prevent loss of the FLAG-tagged mutant? Is viability compromised by insertion of the tag into RL2A (implying the functional polypeptide is critical) or through some effect on the expression of ICP0? The arrangement of the ORFs resembles an upstream overlapping ORF (uoORF) and like other examples might function to regulate translation efficiency of the downstream RL2. The fact that RL2A uses a non-canonical initiation codon (ACG) may be relevant if there is some regulatory connection. Some of these possibilities can be addressed using the isolated transcription unit, independent of the whole virus.

We agree with the reviewer that further validation of RL2A was indicated as RL2 represents a key HSV-1 gene locus. It is well described for HSV-1 that mutations within the repeat regions are not well tolerated and commonly revert very rapidly upon passaging. In the first 3X-FLAG-tagged virus, which we generated, the other (WT) copy of RL2/RL2A remained untouched. This facilitated rapid reversion of our 3X-FLAG-tagged virus to wild-type within 2-3 passages.

To prevent recombination and removal of the 3X-FLAG-tag within this repeat region of our 3X-FLAG-tagged RL2A mutant, we deleted the other wild-type RL2A locus including parts of RL2 from the second repeat by replacing it with an Ampicillin resistance marker. This resulted in the reconstitution of a virus with stable expression of RL2A of the expected size (21.8 kD including the 3XFLAG-tag) detectable by Western blot. Interestingly, this mutant showed substantially reduced (but not complete loss of) ICP0 expression. We subsequently noted that the 3X-FLAG-tag contains an out-of-frame AUG start codon (GATTACAAGG**ATG**ACGACGATAA) in every of the three FLAG-tag repeats. Translation initiation at the respective start codons and thus ribosomes bypassing the ICP0 TaSS, explains the observed loss of ICP0 expression and thereby also the rapid reversion of our primary 3X-FLAG-RL2A mutant. Furthermore, this presumably also explains at least some of the attenuation we observed for the mutant viruses with 3X-FLAG-tagged NTEs, namely for ICP27 and VP5. While these findings validate the expression of RL2A, they highlight the need to carefully consider ectopic translation start site usages when manipulating herpesvirus genomes. We included the data of the new mutant into Fig. 6.

Revising an existing genome annotation – especially one as partisan and entrenched as that of HSV-1 – is always difficult. The authors are to be commended for laying out their logic carefully as a supplementary document. Some transcript isoforms are put aside due to very low abundance. The challenge of validation is a recurring issue in studies advocating expanded viral (or cellular) proteomes and the use of Ribo-seq seems a wise choice. By triple-SILAC whole proteome mass-spec, the authors were able to confirm fewer than 6% of the predicted ORFs. Whether this reflects instability, low expression or both remains a matter of speculation. Overall the manuscript covers a lot of ground and touches only superficially on many interesting and potentially important observations. Of course, there is value to making the data available to broader community in a timely fashion but as the manuscript stands, too much is left requiring further validation or some attempt to explore the biological relevance.

The key aim of this study was to provide an important resource to the field, namely the first comprehensive identification and annotation of HSV-1 gene products. Considering the wealth of information this provides, the biological relevance of this can now be studied.

We would like to point out that we performed extensive validation of the identified viral transcripts using a great variety of powerful high-throughput approaches and bioinformatic analyses. The vast majority of transcription start sites of the novel viral transcripts could thereby be validated at single nucleotide resolution by multiple approaches. Our study thus comprises the most comprehensive characterization and validation of any herpesviral transcriptome to date. Accordingly, the viral transcriptome is of similar complexity. We not only performed standard Ribo-seq experiments but also included time-course experiments using both Harringtonin and Lactimidomycin to enrich for translation start sites. Combined with our novel powerful computational algorithm PRICE (Erhard et al, Nature Methods 2018), this provided reliable information on the viral transcriptome. We demonstrate the accuracy of our data by validating 6 alternative translation start sites of well described viral gene products. We believe that >90% of all novel ORFs and sORFs that we identified will be correct.

The rather poor validation rates for the small viral novel ORFs by mass spec is not surprising as this also holds true for cellular sORF-derived polypeptides. They are thus thought to mainly exert regulatory functions. It will be interesting to clarify whether viral uORFs simply tune or regulate viral gene expression.

SPECIFIC COMMENTS

The new nomenclature uses various symbols to denote extended or truncated transcripts. Will these be clear enough on figures and other maps to be maintained?

The nomenclature which we generated uses “#” – for truncated isoforms and “*” for extended isoforms. Utilizing these symbols for both RNAs and proteins makes it easier to follow and avoids lengthy texts. The two symbols should be readily distinguishable on figures and other maps. The number of viral gene products is so large that it can no longer be comprehensively depicted on a single A4 page. On the contrary, our easy-to-use genome viewer allows visualization of all viral gene products down to single nucleotide resolution. We have already made this available to the broad scientific community. Thereby, any researcher can check by themselves how reliable the data are for an individual viral transcript or ORF. We have already received a lot of positive feedback.

Pg 6 line 208 “Expression of both a 135 kDa (full-length) and 90kDa (NTT) protein has been shown for the commercial ICP8 antibody 11E2 (Santa Cruz, see product website).” There’s a sizeable literature on ICP8 and yet this truncated isoform has not been commented on before. Commercial antibodies are less specific than advertised and it seems odd to cite a non-peer reviewed source. Can this not be tested directly? According to the website image, HSV-1 (MacIntyre strain) infected African Green monkey kidney do not produce the shorter species seen the HSV-1 (17 syn + strain) infected baby hamster kidney cells. Are there clues in the genomic sequence (GenBank accession no. KM222720) as to why this might be?

We agree with the reviewer that this required further independent validation. Unfortunately, we were unable to obtain the respective antibody from Santa Cruz as it is currently out of stock. We thus generated a mutant virus expressing a C-terminally tagged ICP8 (UL29). Interestingly, we could not confirm any expression of the truncated ICP8 (UL29.5) by Western Blot. While we are confident that the truncated form of ICP8 is translated, further studies are required to look at protein stability of the respective protein and its transcript. We carefully rephrased the respective results section (line 213 – 218).

For the most part, the authors limit their analysis to the first 8 hours of infection but the rationale for this and its relationship to major landmarks in the replication cycle (onset of DNA synthesis, appearance of new infectious particles etc) is not laid out.

In our previous study (Rutkowski et al, Nature commun. 2015), we analyzed the HSV-1 induced changes in global transcriptional activity in our experimental setting (HSV-1 infection of HFF, MOI=10) and found this to rapidly drop after 8h p.i. consistent with a global loss of Pol I, II and III transcription later on in infection. The same was observed for candidate genes by qRT-PCR on 4sU-RNA. First virus particles are already released from HFF by 4h p.i. At 8 h p.i., Ribo-seq revealed >80% of translational activity to be viral. We therefore believe that by restricting our analysis to the first 8 h of infection, we cover the majority of the relevant viral transcriptome and translome. Later on in infection, host cell physiology becomes

more and more disrupted. This may well result in the generation of other, noncanonical viral gene products, which are likely to be of questionable relevance to productive infection *in vivo*. We included a sentence on this at the beginning of the discussion.

Unfortunately, I was unable to run the Mac OS version of the genome browser and thus could not assess its utility as a resource.

This is indeed unfortunate and we are sorry for this problem. If hardware/software specifics or any error messages are supplied, we can troubleshoot why this reviewer was unable to run our viewer. Within the last 6 months, we sent our genome viewer to many researchers in the HSV-1 field and did not hear of any major problems. We are happy to help sort this out once contacted.

Some preprints are cited that are now published. These should be updated and consolidated.

We have updated the references per this comment and those below.

Reviewer #2 (Remarks to the Author):

The paper of Whisnant et al. titled 'Integrative functional genomics decodes herpes simplex virus 1' reports a comprehensive analysis of several sets of high-throughput sequencing data of HSV-1 infected cells, leading to comprehensive annotation of HSV-1 transcripts and corresponding ORFs. The main novelty of the paper is the heroic effort to combine multiple types of sequencing data and to integrate it to an accessible resource that can help follow-up genetic studies.

We would like to thank the reviewer for the appreciation of the efforts we took to comprehensively revise the annotation of the HSV-1 genome using all currently available big data sets.

Below are several comments that can help to improve clarity:

1. Data-sets used in the paper: Some of the analyzed datasets were created by new experiment, while others were previously published. Although the novelty of each dataset is stated in the legend of figure 1, throughout the text the distinction between novel and reanalyzed data is not stated clearly enough (in the overview rows 83-90, in the data availability rows 619-627).

We now stated this more clearly in the overview rows 83-90 and the Figure legend of Fig.1 as well as the new supplementary methods.

Additionally, the MinION data is attributed to two separate sources at different places in the text (ref 12 in line 86 and in figure 1 legend line 840, and ref 14 in line 108). State more clearly how the long-read data was used. Currently the writers state regarding the PacBio data that a modified GFF file was used (line 576) with no details for the MinION data. Specifically, Depledge et al. paper (ref 14 and 29) published in February 2019 identified transcription start sites and termination sites using MinION and a very thorough analysis. This publication includes an organized table of all identified transcripts and the ORFs downstream to them. The authors should directly address these previously published results and compare them to the new annotations presented in this manuscript. If they are overlapping they should acknowledge that, if not they should clarify the differences.

We clarified the origin of the data inside the text as well and corrected the references to the MinION data. Here, we would like to clarify, that we did not use the raw MinION and PacBio data, only the transcripts that the respective groups had called. This includes the tables published by Depledge et al in their Nature Communications paper. We further clarified that in the manuscript.

We were able to confirm 78 of the 89 transcripts. The remaining 11 transcripts were not confirmed by any other approach and were therefore not adopted into our final annotation. However, they are not completely lost, as our viewer contains a track for all transcripts called by Depledge et al. The same goes for the transcripts called by the group of Tombacz et al. We now properly clarify this in the manuscript.

Since differential PolyA and read-through is reported, it is important to detail how full transcripts (TiSS to polyA) were annotated (meaning, how it was decided which PAS belongs to which TiSS). The same is true for any alternative splicing events, how were these compiled in to the annotations?

Full length transcripts (TiSS to polyA) were generated by extending all TiSS to the next available polyA site, which were taken from previous studies (47 in total).

Afterwards, all of them were manually curated to check for possible polyA read-throughs and subsequent use of downstream polyA sites, e.g. which was observed for transcripts starting very close to a polyA site. These corrections were done with the help of the transcripts called by Depledge et al. and Tombacz et al. in their MinION and PacBio data, respectively. In some cases (e.g. UL24 PAS) the time-course data showed very clearly that some PAS experienced read-through transcription. We therefore extended the respective transcripts to the next PAS. Splicing events were included in the annotation if they were present at reasonable levels in our total RNA-seq or 4sU-seq data (see Suppl. Fig. 2) or the MinION or PacBio data suggested their presence. The latter were again checked visually. We clarified this in the manuscript.

2. References and previous work:

As indicated above there are two published MinION datasets it is not clear which one was used and how. The only papers cited for sORF function are for uORFs (line 76), some more can be cited for other functional sORFs. Some papers appear twice in the reference: ref 9 and 55, ref 14 and 29 In line 169 the authors cite ref 30 seemingly out of context.

We corrected and clarified the citations of the MinION dataset. We only used the transcripts that were called by the group of Depledge et al. The same goes for the transcripts called by Tombacz in their PacBio dataset. The MinION dataset from Tombacz was not used, as they did not analyze it and we were unable to gather any useful material out of the raw data due to its poor data quality.

We included two recent reviews on sORF-encoded polypeptides.

The reference duplicates were deleted. Ref. 30 was included as part of our data (mock and WT) was already published in this paper. It also includes the respective controls.

3. Splicing:

- the authors show identification of splice junctions but do not specify with which tools where used to detect them

- figure s2 is hard to understand. Are these reads spanning splice junctions? What does upstream/downstream mean in this context?

We used custom scripts (available at zenodo) that counted the number of unique reads spanning the respective splice junctions (red) and compared them to the number of reads that did not splice the junction upstream (green) and downstream (blue). We added a sentence in the manuscript and extended the figure legend for Supplementary Figure 2 to clarify the figure's contents in a more detailed manner.

During the revision of our paper, Tang et al from the Krause lab revealed non-canonical splicing events to arise in HSV-1 infection in absence of ICP27. We now also performed a comparison of these splicing events to our data. Moreover, we tested whether the low-abundance splicing events which we excluded from our new reference annotation might reflect such cryptic splicing events arising in infected cells with insufficient levels of ICP27. However, the vast majority of the 44 splicing events that we observed at low levels in WT HSV-1 infection did not rise in splicing rates (exon1-exon2 to exon1-intron and intron-exon2 ratios). We conclude that insufficient levels of ICP27 in a few of the cells in culture do not explain the rare splicing events we observed. We included a paragraph on this into the manuscript.

4. Data presented in figures

Fig 2:

A- The y axis should be labeled.

Done.

B-C it is unclear why the MinION and PacBio identified TiSS are shown on separate diagrams. Also there are two studies using MinION. Which data was used here? As already clarified in the questions above, the transcripts called in the MinION data by the group of Depledge et al. were used. We added a reference for clarification. Furthermore, two diagrams were used as Venn-diagrams with 4 variables are a lot harder to read. However, our main point here is the correlation of TiSS between our data and the TiSS called by MinION or PacBio respectively. This shows that MinION, although not able to identify a lot of transcripts compared to PacBio, does provide a very high sensitivity, whereas nearly half of the transcripts called by PacBio could not be validated by our methods. This presumably resulted from fragmented viral RNAs that are misidentified as independent transcripts. We edited this part of the manuscript and emphasized more on this point.

D- the criteria of TiSS annotation. It is unclear why the presence of an ORF downstream of an initiation should be a requirement for the annotation of a transcript.

We decided to include this criterion as an ORF essentially requires a transcript to be expressed. This criterion thereby considers that a putative TiSS picked up by cRNA-seq or dRNA-seq is more likely to be real if there is an ORF which is otherwise not explained. In some cases, we cannot exclude the presence of IRES elements to initiate translation of some of these ORFs. However, in most cases, there was simply no other transcription start site downstream of the next upstream poly(A) site.

As poly(A) read-through only occurs relatively late in infection, this was unlikely to provide the required transcripts. Considering the presence of repeat regions and GC-rich sequences, which are known to interfere with the preparation of sequencing libraries, it is not surprising that some transcription start sites were hard to clone and did not provide optimal signals in our datasets. We thus decided to score the presence of an otherwise "orphan" ORF within less than 500nt downstream of a putative TiSS. Based on manual inspection of the respective TiSS, we are confident that this provided a useful criteria to comprehensively annotate the HSV-1 transcriptome. Finally, we would like to point out that our scoring system and the obtained results are included in the list of HSV-1 transcripts and thereby provide researchers with a direct mean to judge the confidence of identification of the respective TiSS.

Fig 3:

A- The difference between the coverage drop after the PAS between the cytoplasm and the other fractions is not very convincing (figure 3a). Some statistic should be added and a gene example to strengthen the point.

We now included the respective statistical analysis to Fig. 3a and have also highlighted the drop in cytoplasmic read levels for the UL30 PAS

B- Since multiple types of RNAseq data were used, it should be specified which one is shown in the genome browser figure 3b. The y axis should be labeled.
Done.

Fig 5:

This figure includes a novel interesting finding, that extension of viral proteins can support different localization. However there are several issues that need to be addressed here:

A-E: Requires explanation as it is strange that the NTE appears only when it is tagged upstream of the main AUG, and not when the main AUG is tagged, since the location of the tag should not affect the expression level. Wouldn't it made more sense to have one tag at the C-terminal that allow assessment of the relative expression by separating the bands. Only in UL54 two bands are seen when the main AUG is tagged but here there is very different intensity compared to the tagging of the NTE, again why? Altogether these raise the concern the expression of the NTE is neglect-able compared to the main AUG. how does this goes along with the measurements of above 10%?

For US3 and UL19, the size difference is too small to distinguish the NTE from the AUG isoform when the AUG isoform is tagged. Unfortunately, for both US5 and UL50, the intensity of the AUG isoform is far too high to see the much weaker expressed NTE. For UL54, we do see the NTE in the AUG-tagged mutant. For the US5 glycoprotein, we also now generated a C-terminally tagged virus. However, in contrast to N-terminally tagged US5 which showed discrete bands, this resulted in a massive smear on the Western blot which precluded interpretations.

Viral gene expression spans 3-4 orders of magnitude between different viral proteins. We restricted calling NTEs to an expression level of at least 10% of the AUG isoform. It needs to be considered that Ribo-seq data reflect the time ribosomes stay on an individual codon before moving on, rather than actual translation rates. Therefore, the 10% value represents a surrogate marker but not an absolute quantification. As the N-terminal extensions are rather short compared to the main ORF, the quantification is relative at best. Nevertheless, there are other known viral proteins which are expressed at lower levels than the respective NTEs. Considering the problem with out-of-frame AUGs in the 3xFLAG-tag, which we identified for RL2A, new tagged viruses in which translation of the main CDS is not affected, will need to be generated to study the function of these N-terminal extension.

F-G: The immunofluorescence of the NTE tagged genes is very weak compared to the AUG tagged ones. This is probably due to expression level differences, however, the weak signal (especially that of US3 NTE) makes it hard to distinguish from background noise. It will be better show it in comparison to background staining (no primary Ab). In addition the preparation of only 2 out of 5 viral mutants are detailed in the methods section. The gene names should be consistent (UL54 in the figure and ICP27 in the main text, etc.)

We now ensure consistency of gene names in the figure and main text. The viral mutants were prepared by the same methods as the WT viruses, which is why only those requiring complementing cells lines are specified. We have now more clearly expressed this in the methods section. We here also included the antibody staining of cells infected with the parental virus lacking a FLAG tag, which was performed in

the same experiment. This demonstrates that, despite relatively lower amounts, the signal is much higher than background noise.

Fig 6:

The different colored frames should be explained clearly (there are 2 types of green). The y axis should be labeled.

C- The expected and observed size should be clearer.

The 2 types of green only resulted from an attempt to superimpose all the data and make them transparent. Each panel only shows one replicate with 3 colors now, one color for each translated frame. We removed the transparency to prevent confusion and added a short sentence to the Figure legend describing which frame is depicted by which color. The y axis was labeled.

We have now also included data on a new 3X-FLAG-tagged RL2A mutant in which the second repeat was replaced by an Ampicillin resistance marker. Both viruses showed RL2A expression of the expected size (21.8 kD). Interestingly, however, ICP0 expression was almost (but not completely) abolished. We subsequently noted that the 3X-FLAG-tag contains an out-of-frame AUG start codon (GATTACAAGG**ATG**ACGACGATAA) in every of the three FLAG-tag repeats. Translation initiation at the respective start codons explains the observed near-complete loss of ICP0 expression and thereby also the rapid recombination of our primary 3X-FLAG-RL2A mutant upon serial passaging. Furthermore, this presumably also explains at least some of the attenuation, which we observed for the mutant viruses with 3X-FLAG-tagged NTEs, namely for ICP27 and VP5. These observations highlight the need to carefully consider ectopic translation start site usages when manipulating herpesvirus genomes.

5. Genome Browser:

The genome browser provided for viewing the data is easy to install and navigate. Some notes: the searching for terms only works when they are at the beginning of the gene names, and the double clicking on genetic element does not work. In addition, easy viewing of data will benefit from adding the option to minimize the size of tracks (such as squished in IGV) or selecting partial tracks out of the main ones (e.g. translation).

We have updated our search-terms to now ignore upper- and lower-case spelling. The double-clicking bug was resolved. The minimization feature is currently under development and intended to be implemented together with the publication of two other genome annotations (HCMV/MCMV). We will then update the HSV-1 viewer accordingly.

Minor comments:

Line 81 "Functional genomics" is used out of context since the annotation here is not of function

Line 141 add "the" before core

Line 186 "selective" should be "selected"

Line 271 delete "only"

We here modified the manuscript accordingly. However, we would like to maintain the term functional genomics as we are looking at transcription and mRNA isoforms explaining translation.

Adding arrows to indicate the strand of current gene will facilitate the understanding of genome browser figures (especially the ones in the supplementary).

We tried this and kindly disagree. For each excerpt of the viewer at the top it is stated on which strand we are on (JN555585+ or JN555585-). This means that ALL depicted transcripts and ORFs are in the specified direction. Adding arrows to transcripts would probably rather confuse the user, as this would seem like there is the possibility of certain transcripts or ORFs being read in the opposite direction.

Reviewer #3 (Remarks to the Author):

The authors of this work have an interesting and 'hot' topic and the paper reads well. However, the actual details and substance are very difficult to follow. It has taken me a long time and after a considerable struggle I am sure that there is no way that I can reproduce their work.

We appreciate the criticism. After reading through it carefully and reiterating over our method section, we have to agree that a lot of the information given did not provide the full picture and was difficult to understand and thus reproduce. Consequently, we have fully rewritten and extended our Supplementary Methods. It now provides a detailed description of our analysis including examples of TiSS, which fulfilled the criteria, from our genome viewer. In addition, we would like to point out that we provide all our source code for the analyses on Zenodo, which do enable the direct reproduction of our work (<https://doi.org/10.5281/zenodo.2621226>). We have now also added additional comments and documentation to these scripts to make them more comprehensible. We are confident, that with the updated method sections and the Zenodo-files, reproducing our results as well as understanding the rationale of the approaches we took has been made as straight-forward as possible.

Further they have a whole new software and the description of the metrics the software uses are cryptic. This is a black box. It is clear the authors have done a lot of work, that is why I initially asked for a full set of methods. The response was somewhat helpful but not nearly detailed enough. I have identified some examples to help guide the authors, but this is far from an exhaustive list. This is so poorly described that it needs a complete rewrite. Someone needs to go slowly step by step and describe 1) how many samples and where they came from 2) the qc metrics 3) the detailed analysis protocols so that these can be reproduced 4) the rationale for each step 5) the detailed raw results for each rep should be provided.

We rewrote and added substantially more content to the Supplementary Methods describing the qc metrics, how many samples were used and the step-by-step analysis that was conducted to call the TiSS for each individual method. This also includes the rationale for each of the algorithms applied.

Further, we updated our files on Zenodo. Each 'start.bash'-file (which is used to start reproducing our analysis and figures denoted by the folder-name in which they are located) now includes a comment-block over each command called, providing additional information about them.

The raw results for each step are already available at Zenodo and can be reproduced by simply using the single 'Makefile' or running them individually with the aforementioned 'start.bash'-files. We hope this now provides all the necessary information to reproduce our work

I am not sure if the MASS Spec Data are not deposited.

The mass spec data were deposited at the PRIDE database referenced in the “data availability” section in our manuscript. The login-credentials for the two datasets are as follows:

PRIDE (PXD013010) Username: reviewer80345@ebi.ac.uk; Password: U74g3ti0

PRIDE (PXD013407) Username: reviewer27776@ebi.ac.uk; Password: X7lvT02V

Characterization of transcriptome, 155 TiSS page 4 line 101, 204 TiSS page 4 line 114. So what is 204? the union of TiSS and dRNA? This type of ambiguity is found throughout the paper.

To avoid ambiguity, we now differentiate between “potential TiSS” and “bona fide TiSS”. A potential TiSS is a single nucleotide position that was identified by iTiSS. A bona fide TiSS is the start of an annotated transcript in our new reference annotation and consists of a single nucleotide position including a +/- 5 bp window around it. In other words, we first identify potential TiSS in the different datasets. Then, we combine them and see if certain positions were called in multiple datasets. While combining, we grant a variability of +/- 5 bp, which then results in our annotated TiSS.

We noted that we had actually mixed up potential TiSS and bona fide TiSS at some point when generating the figures. We therefore highly appreciate the referee’s keen eye for spotting this. We have corrected our analysis and all the numbers stated in Fig. 2 that were affected. However, this only affects the total numbers of potential TiSS for the different approaches. The number of bona fide TiSS is NOT affected.

Line 124 ‘scores for differences’ HOW? The new supplement was very confusing... for example “If the read start count for a positing exceeded an interquartile range of 5, it was considered a potential TiSS and marked as such. If the same position was marked as a potential TiSS in both replicates, the position’s TiSS-score was incremented by one.” What does this mean? That the difference in the 75 and 25 quartiles should be bigger than 5 reads? And then “If the same position was marked as a potential TiSS in both replicates, the position’s TiSS-score was incremented by one.” So what was the start score?

In essence, the “TiSS-score” for a position is equal to the number of criteria that are evidence for transcription initiation and apply at this position. Each criterion is computed in a specific way. Mentioned here is criterion ii (**cRNA-seq read accumulation**). We now provide a much more detailed description of the respective criteria in the Supplementary Methods exemplified for criterion ii (shown below).

ii: cRNA-seq read accumulation: Contrary to the dRNA-seq dataset, cRNA-seq also shows a strong enrichment of reads at the transcription start sites but contains a lot more reads that map to the gene body (~18-fold enrichment of TiSS). The total numbers of reads at the TiSS and within the gene body varied widely akin to the dRNA-seq data set. For this reason, we again chose a moving window approach. To account for reads within the gene body, three moving windows (101 bp each) were used. The first window is located 100 bp downstream of the currently observed position, the second 100 bp upstream and the third +/- 50 bp around it. In HSV-1, multiple transcripts commonly use the same poly(A)-site. Consequently, many transcriptional start sites are located inside another transcript, which started further upstream. Let *a* and *b* be two transcriptional start sites that both utilize the same poly(A)-site with *a* being located upstream of *b*. With cRNA-seq reads mapping throughout the whole transcript, we would expect the read counts of *b* to be greater than the

read levels in between *a* and *b* as well as downstream of *b*. iTiSS accounts for this by employing a widely used outlier filtering approach. The interquartile range, i.e. the difference of read counts of the position at the third quartile and the read counts of the position at the second quartile is calculated for all three windows. Next, the difference between the currently observed position and the third quartile in each window is calculated. If this exceeds a threshold of 5 times the interquartile range within all windows, the position is considered as a potential TiSS. The third window is used as an additional filter to prevent calling potential TiSS in noisy areas of the genome. Those areas usually comprise a significant number of reads accumulating in a ~100 bp region with no reads before and after it. Consequently, around 50% of the first and second window would contain positions with no reads mapping to them, moving the second and third quartile in those windows closer to zero and therefore falsely increasing the number of called potential TiSS. Additionally, if less than 50% of positions in all three windows contained reads mapping to them, the region was disregarded as noise.

“This was done using Fisher’s exact test by defining a threshold (mean read start count downstream) and counting the number of positions that have a lower or higher read start count for the upstream and downstream window, respectively. If there was a significant difference (p -value < 0.01), the position was marked as a potential TiSS.” So Fisher’s exact test is done on a contingency table. I can’t figure out 1) what is being counted in the table and 2) how many millions of tests they have done. For sure a p value of 0.01 is liberal. Correction for multiple testing was applied using the Benjamini-Hochberg correction.

I can’t figure out how many replicates there are of anything except the TiSS which in one line says 2 but the others are not clear- you need to go to the data deposit and there the numbers are disappointing.... what remains unclear is whether these are the same biological samples shared between techniques or are they independent?

Two biological replicates were performed for all high-throughput experiments unless otherwise specified. We included this information in the legend of Fig.1. This information can now also be found in the supplied Data-Source file in tabular form.

The data analysis is completely cryptic. Many of the sentences do not make sense. We have now expanded the methods section to express things more clearly. A manuscript on the iTiSS algorithm is currently in preparation and will provide a much more detailed description of the bioinformatic work. Please note that the scripts to generate all figures and suppl. figures shown in this paper are available at Zenodo with now substantially improved comments and documentation (<https://doi.org/10.5281/zenodo.2621226>). This allows all figures to be rapidly reproduced.

Reviewers' Comments:

Reviewer #1:

Remarks to the Author:

In this revised manuscript, the authors provide the most comprehensive analysis to date of the HSV1 transcriptome by combining a number of different datasets generated from productive infections of primary human fibroblasts (and other human cells) harvested no later than 8 hours post-infection. Studies of HSV1 have led the way in terms of understanding herpesvirus genomes, including the first complete genomic sequence. However, the field has been slow to overhaul the original gene maps, which were based largely on predicted ORFs and similarly slow in embracing NGS approaches. As such, this update brings HSV1 into line with the other major human herpesviruses, HCMV, KSHV and EBV. Hopefully, the new transcript maps will become the lingua franca for future work by the field and will reduce the notorious impenetrability of the HSV1 literature caused by the ill-defined and often redundant existing nomenclature. The new nomenclature preserves the older names avoiding any severe disconnects with prior literature.

Aside from defining the structures of a large number of viral transcripts (5' and 3' ends, boundaries of ORFs and so on), the authors provide evidence in support of 46 new large ORFs (new viral proteins?) and 17 previously unrecognized N terminal extensions in well studied proteins. This data is freely provided to the research community via a custom-built genome browser, which will aid in the adoption of the new annotation by others (although see caveat below).

A chief strength of this integrative approach (using several different RNA-seq methodologies, ribosome profiling and the isolation of both steady-state and nascent RNA and isolation of RNA from different cellular compartments), is that it reduces the false positives that arise when only one methodology is used. There is a good correlation with a recent direct RNA sequencing approach using nanopore arrays (Ref.14) but a less clear relationship to data previously obtained with the PacBio system (Ref.13), most likely reflecting different read count thresholds. The same is true for novel splicing events, with disagreement between different groups with respect to the less abundant splices. The authors make good use of comparisons between WT HSV1 and a mutant virus lacking ICP27, a pleiotropic viral protein active throughout the infection cycle that influences several aspects of mRNA processing and export.

The identification of a number of N-terminal extensions (as well as N-terminal truncations) in canonical proteins is of great interest. Several of these have been experimentally validated and along with similar observations in the literature lead the authors to conclude that 'NTEs initiating from non-AUG start codons are common in alphaherpesvirus proteomes'. This appears to be a frequent mechanism to expand the coding potential or at least to redirect specific viral functions within the infected cell.

Overall this is an interesting and well constructed study. The authors have responded well to the prior reviews and the revised manuscript is more accessible than the original submission. Ultimately this study should provide a valuable resource to the field. Many aspects of the integrative approach can also be applied to other viruses to build, or at least refine, genome annotations. Aside from the technical advances the authors have expanded the known HSV1 proteome in a number of interesting ways. Much work remains including studies of the biology behind the numerous short uORFs encoded within the majority of canonical transcripts and isoforms.

MINOR RECOMMENDATIONS

The annotation browser (available via the Zenodo website) is a useful resource in terms of mining the

extensive data and analysis that is summarized in this manuscript. That said, it is not immediately clear if any users (other than the most computationally sophisticated) will be capable of easily layering in their own RNA-seq data onto this map to allow easy comparisons and further the use of the transcript definitions and nomenclature. Most users are probably more used to the Broad Institute's IGV (Integrative Genomics Viewer) as an interactive visualization tool. Offering an IGV ready version is recommended.

Ln 256. Maybe soften the plausible but as yet untested statement 'US3 NTE contains a leucine-rich stretch indicating a functional nuclear export signal'?

REVIEWER: Angus C Wilson (New York University School of Medicine)

Reviewer #2:

Remarks to the Author:

The authors have fully addressed my comments.

Below are minor comments that might help to increase clarity

Figure 6C- As indicate in the previous round it will be helpful if the expected size/s (in the western blot) will be labeled, or at least some mark of a ladder will be included.

The authors added helpful clarifications for supplementary figure2. It seems that instead of "exon spanning reads" it should be "junction spanning" or "exon-exon junction spanning reads".

The supplementary methods addition was very helpful . A sentence in the description of the Orphan PRF TiSS should probably be rephrased: "The few TiSS, which we did include into our new reference annotation, all showed additional evidence in some of the other criteria, which were only just not called as the respective threshold was missed."

Reviewer #3:

None

Reviewer #1 (Remarks to the Author):

In this revised manuscript, the authors provide the most comprehensive analysis to date of the HSV1 transcriptome by combining a number of different datasets generated from productive infections of primary human fibroblasts (and other human cells) harvested no later than 8 hours post-infection. Studies of HSV1 have led the way in terms of understanding herpesvirus genomes, including the first complete genomic sequence. However, the field has been slow to overhaul the original gene maps, which were based largely on predicted ORFs and similarly slow in embracing NGS approaches. As such, this update brings HSV1 into line with the other major human herpesviruses, HCMV, KSHV and EBV. Hopefully, the new transcript maps will become the lingua franca for future work by the field and will reduce the notorious impenetrability of the HSV1 literature caused by the ill-defined and often redundant existing nomenclature. The new nomenclature preserves the older names avoiding any severe disconnects with prior literature.

Aside from defining the structures of a large number of viral transcripts (5' and 3' ends, boundaries of ORFs and so on), the authors provide evidence in support of 46 new large ORFs (new viral proteins?) and 17 previously unrecognized N terminal extensions in well studied proteins. This data is freely provided to the research community via a custom-built genome browser, which will aid in the adoption of the new annotation by others (although see caveat below).

A chief strength of this integrative approach (using several different RNA-seq methodologies, ribosome profiling and the isolation of both steady-state and nascent RNA and isolation of RNA from different cellular compartments), is that it reduces the false positives that arise when only one methodology is used. There is a good correlation with a recent direct RNA sequencing approach using nanopore arrays (Ref.14) but a less clear relationship to data previously obtained with the PacBio system (Ref.13), most likely reflecting different read count thresholds. The same is true for novel splicing events, with disagreement between different groups with respect to the less abundant splices. The authors make good use of comparisons between WT HSV1 and a mutant virus lacking ICP27, a pleiotropic viral protein active throughout the infection cycle that influences several aspects of mRNA processing and export.

The identification of a number of N-terminal extensions (as well as N-terminal truncations) in canonical proteins is of great interest. Several of these have been experimentally validated and along with similar observations in the literature lead the authors to conclude that 'NTEs initiating from non-AUG start codons are common in alphaherpesvirus proteomes'. This appears to be a frequent mechanism to expand the coding potential or at least to redirect specific viral functions within the infected cell. Overall this is an interesting and well constructed study. The authors have responded well to the prior reviews and the revised manuscript is more accessible than the original submission. Ultimately this study should provide a valuable resource to the field. Many aspects of the integrative approach can also be applied to other viruses to build, or at least refine, genome annotations. Aside from the technical advances the authors have expanded the known HSV1 proteome in a number of interesting ways. Much work remains including studies of the biology behind the numerous short uORFs encoded within the majority of canonical transcripts and isoforms.

We would like to thank the reviewer for his very thorough and positive review.

MINOR RECOMMENDATIONS

The annotation browser (available via the Zenodo website) is a useful resource in terms of mining the extensive data and analysis that is summarized in this manuscript. That said, it is not immediately clear if any users (other than the most computationally sophisticated) will be capable of easily layering in their own RNA-seq data onto this map to allow easy comparisons and further the use of the transcript definitions and nomenclature. Most users are probably more used to the Broad Institute's IGV (Integrative Genomics Viewer) as an interactive visualization tool. Offering an IGV ready version is recommended.

We would like to thank the reviewer for this excellent suggestion. We now compiled all the necessary files to browse our data also in IGV. The BAM-files as well as an "igv-session"-file for the TSS-tracks can be found on Zenodo alongside the other scripts.

Ln 256. Maybe soften the plausible but as yet untested statement 'US3 NTE contains a leucine-rich stretch indicating a functional nuclear export signal'?

We replaced "functional" by "putative". We agree that this is more appropriate.

REVIEWER: Angus C Wilson (New York University School of Medicine)

Reviewer #2 (Remarks to the Author):

The authors have fully addressed my comments.

Below are minor comments that might help to increase clarity

We would like to thank the reviewer for the very thorough and positive review.

Figure 6C- As indicate in the previous round it will be helpful if the expected size/s (in the western blot) will be labeled, or at least some mark of a ladder will be included.

We now include the respective size marker information to Fig. 6c.

The authors added helpful clarifications for supplementary figure2. It seems that instead of "exon spanning reads" it should be "junction spanning" or "exon-exon junction spanning reads".

We changed it to "Junction spanning".

The supplementary methods addition was very helpful . A sentence in the description of the Orphan PRF TiSS should probably be rephrased: "The few TiSS, which we did include into our new reference annotation, all showed additional evidence in some of the other criteria, which were only just not called as the respective threshold was missed."

We removed this sentence as it was redundant. We now describe in much more detail how the manual curation was performed. This includes a section on "Manual curation" in the Suppl. Methods. Moreover, we included a brief description why the respective TiSS were included into Suppl Tab. 1, column U.

Reviewer #3:

* The current manuscript doesn't provide sufficient information on the iTiSS approach to evaluate it and the approach isn't peer-reviewed and/or published yet. A revised manuscript would need to provide all information for our reviewers to check the validity and performance of iTiSS. Please also take into consideration that we would not be able to publish the manuscript (should reviewers find it suitable for publication) without this information being available.

We appreciate the reviewer's recommendation. However, we would like to note that for validating the general performance of any such method, a gold standard to evaluate against is necessary. For calling transcription start sites (in particular in a small and heavily transcribed genome), there is no proper gold standard (see below). Furthermore, the focus of this manuscript was not to propose the best general method for calling TiSS, but to compile a comprehensive and accurate annotation for HSV-1. We believe that the best approach to validate this is to integrate many large-scale data sets. Thus, the other criteria included that are independent of iTiSS (e.g. 3rd generation sequencing, existence of an unexplained uORF, ...) are an internal validation of the performance of iTiSS for HSV-1. Nevertheless, we now performed extensive additional validation of the iTiSS pipeline.

In particular, we used the current annotation of the human genome and the datasets provided by the FANTOM5-project to estimate the positive predictive value (80.2%) and sensitivity (71.0%) of iTiSS, respectively for cellular genes (all scripts are publicly available on Zenodo). We would like to note that both the Ensembl annotation and FANTOM5 data should not be considered gold standards and include false positives as well as false negatives (i.e. both performance estimates are lower bounds). Nevertheless, we are confident that our HSV-1 genome annotation, which considers all currently available big data, provides a very reliable and valuable resource to the field.

We previously mentioned a "manuscript in preparation" for iTiSS in the last version of this manuscript. This referred to a general application of iTiSS to public data sets. We now decided against this and rather expanded on iTiSS in the Suppl. Methods section of this manuscript to deal with this reviewer's comments. Additionally, we now provide the source code of iTiSS on Zenodo alongside the other scripts.

Finally, we would like to point out that iTiSS is not a single complicated algorithm (black box), which identifies viral TiSS. In contrast, it simply analyses the available 2nd and 3rd generation sequencing data for a total of 9 criteria to screen for potential viral TiSS. The respective criteria are explained in detail in the Suppl. Methods (now including examples). iTiSS then provides a list of potential viral TiSS and the respective scores obtained for them. Upon manual inspection of all potential viral TiSS, we decided to include all TiSS with a score >3 into our final reference annotation. Furthermore, potential

TiSS with a score of 2 scored by criterion (i)-(viii) were also considered to be bona-fide TiSS. Upon careful manual inspection they were all found to be highly likely to represent bona-fide TiSS.

* Please provide more details on methods. For example, it seems that the information on zenodo allows to reproduce figures but details necessary to reproduce the analysis (that is how exactly were the data processed to get to the results that are then plotted) appear to be missing on zenodo or in methods.

The integrative analysis of many large-scale data sets is a complex process that involves processing of each individual data set as well as all the steps integrating them. We placed a lot of effort into making the details of our analysis as clear as possible. We described all details necessary to reproduce all our analyses available in our extensive methods part of the manuscript. In addition, the archive on Zenodo provides an executable form of this description (all figures can be reproduced by starting shell scripts). All these steps are extensively documented (e.g. comments in the shell scripts about the details). We improved the README to make it easier to navigate to the corresponding folders containing the respective analysis part. We feel confident that all this information enables researchers to fully reproduce all computational analyses we performed.

* Please provide more details in methods and be more accurate. For example statements like "only slightly missed" and "we manually screened" should be avoided and instead sufficient accurate details should be provided.

We agree with the referee that these formulations were too vague. We now included a section on "Manual curation" in the Supplementary Methods.

Once the automatic scoring was done, potential TiSS with a score >3 were accepted into our final annotation following manual inspection. Furthermore, potential TiSS with a score of 2 scored by criterion (i)-(viii) were also considered bona-fide TiSS. Nevertheless, they were all carefully inspected manually and all found to highly likely represent bona fide TiSS. All remaining potential TiSS were manually curated by looking at the data in our viewer. In particular, we consider the orphan ORF TiSS criterion (ix) the weakest piece of evidence for a bona-fide TiSS. For this reason, we removed TiSS that only fulfilled this and one other criterion, and only kept those that exhibited additional strong evidence (for instance a fold-change between 3 and 4 instead of the picked threshold of 4 in dRNA-seq). In addition, we had a close look at the nucleotide sequence at the TiSS looking for factors that could have impeded cloning or mapping of the respective reads, e.g. poly(C) or poly(G) stretches as well as repeat regions.

Information on the reasons for including each respective potential TiSS into the final HSV-1 genome annotation are included in Suppl. Tab. 1, column U. This is now exemplified for UL36.4 mRNA

Reviewers' Comments:

Reviewer #4:

Remarks to the Author:

The authors present a deep analysis of the HSV-1 transcriptome during the first 8 hours of lytic infection. They used multiple RNA library protocols that allowed them to detect novel transcripts and ORFs, such as: total RNA, cRNA-seq, dRNA-seq, 4sU-seq and Ribo-seq. In addition, they performed proteomic analysis and compared their findings to published long reads outputs.

They annotated 201 viral transcripts and 284 ORFs. This comprehensive work extends existing knowledge of HSV-1 annotation and therefore is highly valuable.

General Remark

The authors performed a time course experiment using some of the RNA library protocols. Yet, they do not present a time course analysis of the 201 viral transcripts or ORFs.

Specific Remarks

1. Determining viral TiSS candidates

Initially the authors describe a two out of four (cRNA-Seq, dRNA-Seq, MinION and Pacbio) approach that identifies 102 TiSS. In this they do not explain how they identify the TiSS from the cRNA-Seq and dRNA-Seq data. Need to point out where in the article it is explained.

The next paragraph goes on to explain the 9 criteria to identify 189 TiSS from candidates identified by only one method of the four above, yet if my understanding is correct it contains also the 102 TiSS described in the paragraph above.

My understanding is that these 189 are the total count and therefore I would expect a sentence to summarize both paragraphs of the TiSS analysis (lines 97-147). Furthermore perhaps the initial paragraph is not required.

2. Figure 1

The location of the cells used i.e. mock, wt ... within the figure is confusing. I suggest to put it below the line of the RNA fractions.

3. TATA box

Line 152 – for weakly expressed RNA the TATA motifs were rarely observed ($p < 10^{-6}$).

What is p value representing the presence of TATA in the highly expressed or in the low expressed? Which time point and RNA data was used to build the expression categories?

4. Splicing events

Line 169- Regarding the

I found that the junctions

JN555585+:79883-80087 UL36.6_RNA

JN555585+:79883-80090 UL36.6_RNA

Might not be a real junction but problematic mapping with soft clipping.

I therefore suggest to go over the junctions presented and filter them as needed.

5. Figure 2F

Define export index

6. Figure 3a

Analysis of the 500nt downstream of viral polyA, can be problematic when the termination site of one gene such as UL36 is near the terminations of another gene on the opposite strand. Therefore, explain whether the strand information was used during quantification.

7. Figure 3b

Why was the cRNA-Seq selected to be shown and not a more suitable protocol to capture the ends of transcripts such as total RNA?

Within the genomics region selected there are additional genes UL31-UL32 that are not depicted. Please add it unless there is a real reason to omit it. Also, add the 5'-3' orientation of all the genes within the genomic region in this figure.

Are the authors certain that the coverage shown is not from reads belonging to other genes?

8. Methods

Line 522 - For the various RNA-Seq sets indicate read yield.

Line 596 - Data Analysis section

Reads were mapped to human genome, human transcriptome and HSV-1.

Please indicate if for analysis of HSV-1 the reads used were those that aligned only to HSV-1.

Please provide statistics on mapping to each of the references.

Lines 602 - 604 needs to be rephrased. Not clear what is minimum mismatches and what is paying attention to sequencing errors?

9. Annotation

The IGV files you provide are a very important resource. Yet, a very helpful file that you should add is a gtf file that reflects the complete set of transcripts and ORFs, as well as all TiSS defined.

Relevant to junctions –see the junction reads contain clipped region

Response to Reviewer #4

Reviewer #4 (Remarks to the Author):

The authors present a deep analysis of the HSV-1 transcriptome during the first 8 hours of lytic infection. They used multiple RNA library protocols that allowed them to detect novel transcripts and ORFs, such as: total RNA, cRNA-seq, dRNA-seq, 4sU-seq and Ribo-seq. In addition, they performed proteomic analysis and compared their findings to published long reads outputs. They annotated 201 viral transcripts and 284 ORFs. This comprehensive work extends existing knowledge of HSV-1 annotation and therefore is highly valuable.

General Remark

The authors performed a time course experiment using some of the RNA library protocols. Yet, they do not present a time course analysis of the 201 viral transcripts or ORFs.

We indeed did not perform this for the following reasons:

Due to the extensive overlap of the viral transcripts with multiple transcripts commonly terminating at the same poly(A) site, it is not possible to properly differentiate the kinetics of the individual transcripts with standard cRNA-seq or dRNA-seq, even when combined with chemical inhibitors. We are currently establishing the combination of dRNA-seq and SLAM-seq, which will overcome this problem.

Based on the available Ribo-seq data, we cannot reliably differentiate immediate early, early and late viral genes for the viral ORFs. Viral DNA replication already starts at 2hpi with viral late proteins becoming detectable. Further work will be required to elucidate the molecular mechanism governing viral late gene expression.

Specific Remarks

1. Determining viral TiSS candidates

Initially the authors describe a two out of four (cRNA-Seq, dRNA-Seq, MinION and Pacbio) approach that identifies 102 TiSS. In this they do not explain how they identify the TiSS from the cRNA-Seq and dRNA-Seq data. Need to point out where in the article it is explained.

TiSS in dRNA-seq and cRNA-seq were called as described in criteria i-vi. We now made this clear in the text by referring to the description of this in our Suppl.Methods.

The next paragraph goes on to explain the 9 criteria to identify 189 TiSS from candidates identified by only one method of the four above, yet if my understanding is correct it contains also the 102 TiSS described in the paragraph above.

My understanding is that these 189 are the total count and therefore I would expect a sentence to summarize both paragraphs of the TiSS analysis (lines 97-147). Furthermore perhaps the initial paragraph is not required.

The understanding of this reviewer regarding the 102 and 189 called TiSS, respectively is absolutely correct. We took his advice and deleted the sentence containing the 102 TiSS as we agree that it can cause confusion and does not serve any further purpose.

2. Figure 1

The location of the cells used i.e. mock, wt ... within the figure is confusing. I suggest to put it below the line of the RNA fractions.

We updated the Figure according to the reviewer's suggestions

3. TATA box

Line 152 - for weakly expressed RNA the TATA motifs were rarely observed ($p < 10^{-6}$).

What is p value representing the presence of TATA in the highly expressed or in the low expressed? Which time point and RNA data was used to build the expression categories?

The p-value was derived by categorizing the viral RNAs into three groups (low, mid and high transcription). We then checked for a TATA-box or TATA-box like motif 20-25bp upstream of the TiSS. We used Fisher's exact test to identify significant differences between low and high transcription in terms of the presence of TATA-boxes. We agree that this was not described sufficiently. We added more details to the Methods section and two sentences to the main text for a better understanding.

4. Splicing events

Line 169- Regarding the I found that the junctions

JN555585+:79883-80087 UL36.6_RNA

JN555585+:79883-80090 UL36.6_RNA

Might not be a real junction but problematic mapping with soft clipping.

I therefore suggest to go over the junctions presented and filter them as needed.

We now further tightened our rules for which reads to consider by removing reads that contain mismatches around the splice-site. Still, all splice-junctions currently included in our reference annotation occurred at the same or nearly the same level. Reads might be soft-clipped at the 5' or 3' end. However, soft-clipping per-se cannot introduce spurious exon intron boundaries. Importantly, a read is only considered to span a splice junction if there are at least 10 bases mapped on each side of the junction. The aforementioned NAGNAG splice junctions are backed by 120 and 170 reads. We updated our Supplementary Figure 2 with the tightened rules.

5. Figure 2F

Define export index

The export index is defined by the log fold-change between cytoplasmic and chromatin-associated FPKM-normalized read counts. We added this information to the Figure's caption.

6. Figure 3a

Analysis of the 500nt downstream of viral polyA, can be problematic when the termination site of one gene such as UL36 is near the terminations of another gene on the opposite strand. Therefore, explain whether the strand information was used during quantification.

Libraries were prepared using the TruSeq Stranded Total RNA kit from Illumina, which provides strand-specific data. Consequently, genes ending in close proximity on different strands do not result in problems in our analysis. We added a sentence in the data analysis section that strand information was always considered and used.

7. Figure 3b

Why was the cRNA-Seq selected to be shown and not a more suitable protocol to capture the ends of transcripts such as total RNA?

We agree with the reviewer, that our choice of cRNA-seq for this figure was suboptimal. We took his/her advice and now show the 4sU-RNA instead as this represents new RNA and thus changes in transcriptional activity and RNA processing.

Within the genomics region selected there are additional genes UL31-UL32 that are not depicted. Please add it unless there is a real reason to omit it. Also, add the 5'-3' orientation of all the genes within the genomic region in this figure.

The omitted genes (UL31 & UL32) are expressed from the other strand. In this figure, we only depict the plus-strand. We now included a statement on this in the figure legend. As described above, our sequencing protocols are strand-sensitive and showing the antisense strand would require a lot of space but would not add useful information.

Are the authors certain that the coverage shown is not from reads belonging to other genes?

Absolutely, with strand-specific protocols the chances of falsely accumulating reads antisense is very minor and should only be considered if there are >100x more reads on one strand than on the other, which is not the case in this region (see attached genome browser screenshot. Read levels of the plus-strand(top) and minus-strand(bottom) are at similar levels in their respective gene regions. Read levels are marked in red.).

8. Methods

Line 522 - For the various RNA-Seq sets indicate read yield.

Line 596 - Data Analysis section

Reads were mapped to human genome, human transcriptome and HSV-1.

Please indicate if for analysis of HSV-1 the reads used were those that aligned only to HSV-1.

Please provide statistics on mapping to each of the references.

We created a separate Supp. Table holding the read yields and mapping statistics (in total and for each reference) and referred to it in the manuscript. Before all other analyses, read mappings to genome/transcriptome/HSV-1 were merged into a single mapping (discarding mappings suboptimal in terms of mismatches). Only a negligible number of reads (n=432, ~0.001% (dRNA-seq), n=~11k, ~0.001% (cRNA-seq), n=25k, ~0.0003% (4sU & total RNA-seq)) mapped to more than one location. These were scattered across the whole HSV-1 genome and did therefore not interfere with our analysis. Statistics (e.g. multi-mapping/uniquely mapping reads) and a more detailed description of read mapping and processing were added to the manuscript.

Lines 602 - 604 needs to be rephrased. Not clear what is minimum mismatches and what is paying attention to sequencing errors?

We added the following more descriptive statement about our cRNA-seq analysis to the manuscript:

The HSV-1 genome consists of two components (L and S) that are both flanked by long repeats. To mitigate the effect of multi-mapping reads, we masked the terminal repeats by NNN. The three mappings were merged and only the alignments for a read with minimal number of mismatches were retained. Reads were assigned to their specific samples based on the sample barcode. Barcodes not matching any sample specific sequence were removed. PCR duplicates of reads mapped to the same genomic location were identified by counting UMIs. If two observed UMI differed by only a single base, one likely is due to a sequencing error. Thus, we discarded one of the two in such cases. If the reads at this location mapped to k locations (i.e. multi-mapping reads for k>1), a fractional UMI count of 1/k was used. Finally, all read mappings in the repeats were copied into the previously masked regions.

9. Annotation

The IGV files you provide are a very important resource. Yet, a very helpful file that you should add is a gtf file that reflects the complete set of transcripts and ORFs, as well as all TiSS defined.

We now provide a gtf-version of the HSV-1 annotation alongside the GenBank file containing all transcripts (with their respective identified TiSS) and ORFs at Zenodo (doi: <https://doi.org/10.5281/zenodo.2621226>).